# HCN channels at the cell soma ensure the rapid electrical reactivity of fast-spiking interneurons in human neocortex

**Viktor Szegedi**[1,2☯], **Emőke Bakos**[1,2☯], **Szabina Furdan**[1,2], **Bálint H. Kovács**[3], **Dániel Varga**[3], **Miklós Erdélyi**[3], **Pál Barzó**[4], **Attila Szücs**[2,5], **Gábor Tamás**[6], **Karri Lamsa**[1,2]*

**1** Department of Physiology, Anatomy and Neuroscience, University of Szeged, Szeged, Hungary,
**2** Hungarian Centre of Excellence for Molecular Medicine Research Group for Human neuron physiology and therapy, Szeged, Hungary, **3** Department of Optics and Quantum Electronics, University of Szeged, Szeged, Hungary, **4** Department of Neurosurgery, University of Szeged, Szeged, Hungary, **5** Neuronal Cell Biology Research Group, Eötvös Loránd University, Budapest, Budapest, Hungary, **6** MTA-SZTE Research Group for Cortical Microcircuits, Department of Physiology, Anatomy and Neuroscience, University of Szeged, Szeged, Hungary

☯ These authors contributed equally to this work.
* klamsa@bio.u-szeged.hu, karri.lamsa@hcemm.eu

**Data Availability Statement:** All relevant data are within the paper and its Supporting Information files.

## Abstract

Accumulating evidence indicates that there are substantial species differences in the properties of mammalian neurons, yet theories on circuit activity and information processing in the human brain are based heavily on results obtained from rodents and other experimental animals. This knowledge gap may be particularly important for understanding the neocortex, the brain area responsible for the most complex neuronal operations and showing the greatest evolutionary divergence. Here, we examined differences in the electrophysiological properties of human and mouse fast-spiking GABAergic basket cells, among the most abundant inhibitory interneurons in cortex. Analyses of membrane potential responses to current input, pharmacologically isolated somatic leak currents, isolated soma outside-out patch recordings, and immunohistochemical staining revealed that human neocortical basket cells abundantly express hyperpolarization-activated cyclic nucleotide-gated cation (HCN) channel isoforms HCN1 and HCN2 at the cell soma membrane, whereas these channels are sparse at the rodent basket cell soma membrane. Antagonist experiments showed that HCN channels in human neurons contribute to the resting membrane potential and cell excitability at the cell soma, accelerate somatic membrane potential kinetics, and shorten the lag between excitatory postsynaptic potentials and action potential generation. These effects are important because the soma of human fast-spiking neurons without HCN channels exhibit low persistent ion leak and slow membrane potential kinetics, compared with mouse fast-spiking neurons. HCN channels speed up human cell membrane potential kinetics and help attain an input–output rate close to that of rodent cells. Computational modeling demonstrated that HCN channel activity at the human fast-spiking cell soma membrane is sufficient to accelerate the input–output function as observed in cell recordings. Thus, human and mouse fast-spiking neurons exhibit functionally significant differences in ion channel composition at the cell soma membrane to set the speed and fidelity of their input–

**Funding:** Project no. TKP-2021-EGA-05 has been implemented with the support provided by the Ministry of Culture and Innovation of Hungary from the National Research, Development and Innovation Fund, financed under the TKP2021-EGA funding scheme (KL). The project has received funding from the EU's Horizon 2020 research and innovation program under grant agreement No. 739593 (KL). In addition, this work was supported by Nemzeti Kutatási, Fejlesztési és Innovaciós Alap, OTKA K 134279 (VS, SF, KL), Magyar Tudományos Akadémia - the National Brain Research Program Hungary (VS, and KL) and by University of Szeged Open Access Fund (Grant number: 4373). The funders had no role in study design, data collection and analysis, decision to publish, or preparation of the manuscript.

**Competing interests:** The authors have declared that no competing interests exist.

**Abbreviations:** ANOVA, analysis of variance; dSTORM, direct stochastic optical reconstruction microscopy; EPSC, excitatory synaptic current; EPSP, excitatory postsynaptic potential; HCN, hyperpolarization-activated cyclic nucleotide–gated; IQR, interquartile range; pv, parvalbumin.

output function. These HCN channels ensure fast electrical reactivity of fast-spiking cells in human neocortex.

## Introduction

Fast-spiking GABAergic interneurons of the mammalian cerebral cortex are critical for neural information processing as they deliver precisely timed inhibitory outputs that regulate neural networks [1–3]. Precise operation of these neurons relies on "fast in–fast out" properties that allow transformation of excitatory synaptic inputs into inhibitory outputs with high fidelity, short latency, and submillisecond temporal precision (see [4]). Fidelity, time-lag, and jitter in the input–output function arise mainly during the transformation of excitatory postsynaptic potentials (EPSPs) into action potential output [4–6]. This EPSP–spike transformation occurs at or near the cell soma through a process of synaptic integration involving both passive current spread and active (voltage-dependent) amplification that is well described experimentally and theoretically in various rodent neuron types, including fast-spiking GABAergic inhibitory neurons [1,7,8]. Although studies on human neurons have revealed "human-specific" features in synaptic EPSPs as well as action potential firing compared with corresponding animal neurons [9–17], it is poorly known whether human cortical interneurons demonstrate species-specific electrophysiological features and how these influence the "fast in–fast out" transformation of EPSPs into action potential output.

Direct experimental investigation of human neurons is essential for understanding complex cortical function because there are certain minor differences in morphology, synaptic input distribution, and ion channel expression profile between analogous neuronal types in human and rodent cortex [10,14,17–25], and even minor divergence between neurons can result in distinct neuronal network-level responses [26–28]. Although transcriptomic studies have found relatively conserved gene expression patterns among cortical GABAergic inhibitory neurons, species-specific specializations are also apparent [29–32]. Indeed, analogous neuronal types such as parvalbumin (pv)-expressing fast-spiking interneurons show interspecies differences in neurite arborization [33], ion channel expression pattern [23,34], and kinetics of synaptic transmission [18,35]. In general, mammalian pv interneurons exhibit rapid membrane potential kinetics with fast high-fidelity action potential generation [11,36–40], but these electrophysiological properties differ quantitatively between species [35,38,40,41]. These differences could be explained by variations in the densities or membrane distributions of voltage-sensitive ion channels and by distinct passive electrical membrane properties (such as cell membrane persistent ion leakage). However, the details of these parameters in human inhibitory neurons are largely unknown.

Excitability in some neurons is enhanced by hyperpolarization-activated cyclic nucleotide–gated (HCN) channels, which open at membrane potentials from negative to −50 mV and are permeable to $K^+$ and $Na^+$, thereby causing a positive shift in cell resting potential and allowing faster membrane potential kinetics and more sensitive EPSP–spike coupling at the cell soma [21,42–45]. RNA sequencing studies and proteome analyses show high levels of HCN channel isoforms HCN1 and HCN2 in the human and rodent neocortex [21,46]. However, recent electrophysiological studies utilizing surgically resected human brain tissue have shown that HCN channel activity in human neocortex excitatory neurons is much stronger than in their mouse analogs compared to their mouse analogues, and this difference is particularly large at the cell soma [21,24,47]. In rodent pv interneurons, HCN channels are predominantly expressed at the axon [46,48] where they remotely regulate firing frequency and

neurotransmitter release [49–51]; however, they are largely or completely absent from the pv cell soma in the neocortex [40–42,52–55].

In the present study, we examined whether there are differences in HCN channel distribution and function between inhibitory neurons of the same brain region across species and whether these differences have a "human-specific" impact on inhibitory neuron function. To address these questions, we investigated the density of soma membrane HCN1 and HCN2 channels on human pv neurons from surgical specimens of frontal and temporal lobes using immunofluorescence and high-resolution direct stochastic optical reconstruction microscopy (dSTORM) [56], compared electrophysiological signs of HCN channel activity between these human pv neurons and mouse pv neurons, and examined the effects of HCN channel blockade on resting membrane potential and EPSP-to-spike fidelity and speed. We restricted these comparisons to interneurons of the upper supragranular layer, specifically to layers 2 to 3 (L2/3), as this region is associated with higher-order sensory and cognitive signal processing and shows profound evolutionary divergence between humans and other mammalian species [57–61].

Results reveal that the activity of somatic HCN channels in human fast-spiking interneurons accelerate the otherwise relatively slow membrane potential responsiveness, leading to an EPSP–spike transformation rate nearly as fast as that of rodent neocortical interneurons. Considering the vast number of neurons in the neocortex, the observed millisecond-range enhancement in input–output speed at individual neurons may have a large impact on the overall computational speed of large neocortical neuronal networks.

## Results

We performed somatic whole-cell and outside-out patch-clamp recordings of fast-spiking interneurons in L2/3 of human neocortex to assess HCN channel-mediated activity and the influence of this activity on input–output functions, including EPSP–spike transformation. Brain tissue was obtained from patients receiving surgery for deep-brain pathological targets and then sectioned for brain slice recordings. Most recordings were performed in sections of frontal cortex or temporal cortex, but a few neurons were recorded from other regions of neocortex. Patient data, neuron fast-spiking properties, and results of pv immunostaining are summarized in S1 Table. For comparison, we also examined fast-spiking pv-expressing neurons in mouse frontal and parietal cortex.

### Most fast-spiking basket cells in human but not mouse neocortex exhibit a robust somatic sag potential mediated by HCN channels

Whole-cell current-clamp recordings from human fast-spiking cell somata (Fig 1A1) revealed a robust membrane potential sag at the onset of hyperpolarization to −90 mV from −70 mV (step duration, 250 to 500 ms) (Fig 1A2). Cells ($n = 72$) were identified as fast-spiking interneurons by their narrow action potential and extremely modest firing frequency accommodation [35,62]. Further, 58 of these cells were successfully visualized and identified morphologically as basket cells [35]. Of these, 43 were conclusively demonstrated to express the basket cell marker protein pv, whereas pv immunostaining was unsuccessful or inconclusive in the remaining 15 visualized cells. Fourteen cells lacking anatomical visualization were identified as fast-spiking interneurons based on electrophysiological properties (fast action potential kinetics and negligible firing frequency accommodation at high firing rate) [35,62]. In contrast to human fast-spiking cells, similar voltage steps failed to evoke a substantial somatic hyperpolarization-induced membrane potential sag in most mouse fast-spiking pv cells (Fig 1B1–1B2).

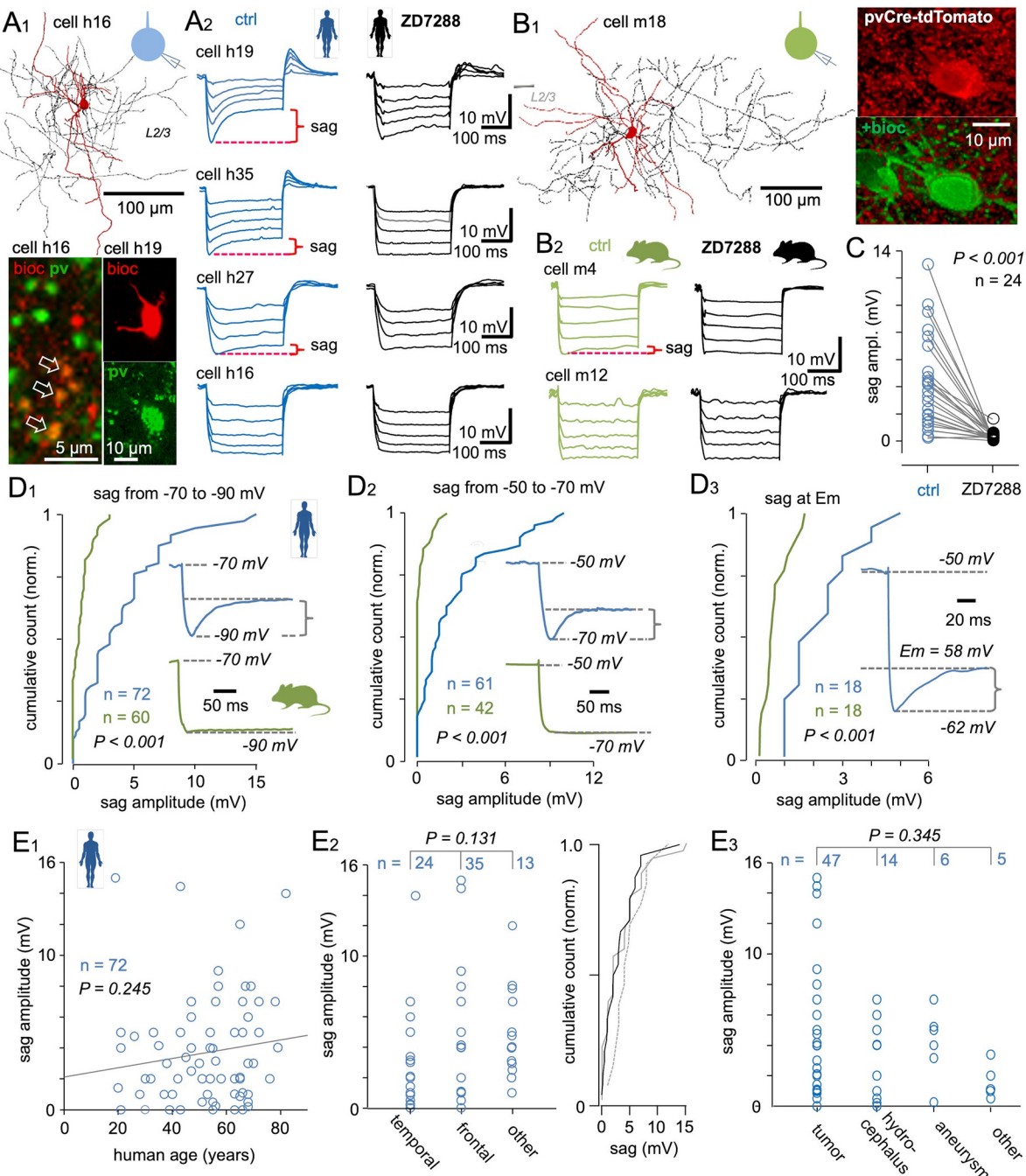

**Fig 1. Human but not mouse fast-spiking basket cells exhibit a robust somatic HCN channel-mediated sag potential.** (**A**) Somatic sag potentials recorded from human fast-spiking pv interneurons. (**A1**) Anatomical and immunohistochemical illustrations of human fast-spiking pv-immunopositive basket cells filled with biocytin during whole-cell recording from the soma. (Inset schematics shows whole-cell clamp protocol.) Top: Partial anatomical reconstruction showing the soma and dendrites (red) and axon (black) of cell h16 (visualized in two 60-μm thick merged sections). L2/3: neocortical layer 2–3. Bottom: Confocal images showing pv immunofluorescence in a bioc-filled neuronal axon (cell h16, pv, and bioc merged, costained axon boutons indicated by arrows) and a neuronal soma (cell h19). (**A2**) Human fast-spiking basket cell membrane potential responses to hyperpolarizing square-pulse current steps delivered from −70 mV. Traces reaching a peak hyperpolarized potential of −90 mV illustrate the hyperpolarization-activated sag potential in three of four sample neurons under control conditions (blue traces). Sample recordings show cell h19 with large sag, cells h35 and h27 with close to median-sized sag, and a cell without sag (cell h16). Black traces show responses of these same cells in the presence of the HCN channel blocker ZD7288 (ZD, 10 μM). Red markings indicate the ZD-sensitive sag potential amplitude (sag) measured during a hyperpolarizing step from −70 mV to −90 mV. (**B**) Absence of robust somatic sag potential in mouse fast-spiking interneurons. (**B1**) Left: Partial reconstruction of a mouse fast-spiking pv-expressing cell filled with bioc during whole-cell recording (inset schematic shows whole-cell clamp protocol). Right: Same cell (m18) with

fluorescence signal (pvCre-tdTomato) indicating pv expression and visualized with biocytin (+bioc). (**B2**) Sample membrane potential traces from two mouse fast-spiking pv-expressing basket cells (control conditions, green; in ZD, black) showing a small voltage sag in one (cell m4) but no sag in the other (cell m12). The latter cell is typical of most fast-spiking mouse neurons recorded. (**C**) The robust somatic sag potential is mediated by HCN channels. Plot summarizes the sag amplitude (ordinate) in 24 fast-spiking human basket cells measured by a membrane potential step to −90 mV from −70 mV under control conditions. Sag potential is blocked by wash-in of ZD ($P < 0.001$ by Wilcoxon signed-rank test). (**D**) Somatic HCN sag potential is common and robust in human fast-spiking basket cells compared to corresponding mouse cells. (**D1**) Cumulative histogram showing HCN sag potential amplitudes in human (blue) and mouse (green) basket cells evoked by a hyperpolarizing somatic step from −70 mV to −90 mV. The HCN sag amplitude is larger in human cells ($n = 72$) than mouse cells (green, $n = 60$) ($P < 0.001$ by Mann–Whitney U test). Sample cells with average of five voltage traces are illustrated in the inset. (**D2**) Histogram of HCN sag amplitude in human and mouse basket cells evoked by a membrane potential step from −50 mV to −70 mV. The HCN sag amplitude in human basket cells is larger than in mouse basket cells ($P < 0.001$ by Mann–Whitney U test). Insets show sample voltage traces from a human and a mouse basket cell. (**D3**) Histogram of HCN sag amplitude measured at the resting membrane potential (Em) following a depolarizing step to −50 mV. The depolarizing step deactivates HCN channels, while repolarization to Em activates these channels, generating a sag potential as illustrated in the inset. The HCN deactivation–activation voltage step cycle applied at Em revealed larger sag potentials in human than mouse basket cells ($P < 0.001$ by Mann–Whitney U test), although Em values did not differ significantly between species ($P = 0.09$ by Mann–Whitney U test). (**E**) Somatic sag potential is a general feature of human fast-spiking basket cells in the neocortex. (**E1**) The sag potential in human fast-spiking basket cells is observed at all ages from 20 to 82 years, and sag amplitude shows no significant correlation with patient age (R = 0.139, $P = 0.245$ by Pearson's correlation analysis). (**E2**) Robust sag potentials in fast-spiking basket cells from different neocortical areas. Left: Plot showing sag potential amplitudes of cells from temporal ($n = 24$), frontal ($n = 35$), and other ($n = 13$) cortical regions. "Other" cortices include occipital, parietal, ventral, and periventricular areas. There is no difference in sag potential amplitude between areas ($P = 0.131$ by ANOVA on ranks). Right: Cumulative histograms of these same data (temporal, black; frontal, gray solid; other, gray dotted line). (**E3**) Prominent sag potentials are also observed in cells from neocortical tissue samples resected due to different primary diagnoses, and amplitudes do not differ among tissue samples resected for tumor, hydrocephalous, or aneurysm treatment ($P = 0.345$ by ANOVA on ranks). The underlying data supporting Fig 1C–1E can be found in S1 Data. ANOVA, analysis of variance; bioc, biocytin; HCN, hyperpolarization-activated cyclic nucleotide–gated; pv, parvalbumin.

The voltage sag in human fast-spiking interneurons was blocked by the HCN channel antagonist ZD7288 (10 μM) ($P < 0.001$ using Wilcoxon signed-rank test, $n = 24$) (Fig 1C). In addition, drug infusion (3 to 5 min) was associated with a significant hyperpolarizing shift from a median resting potential (Em) of −62.0 mV (IQR, −55 to −67.25 mV) to −68.75 mV (IQR, −56.50 to −74.25 mV), for a median shift of −6.75 mV (IQR, −2.8 to 9.0 mV; $P = 0.004$ by Wilcoxon signed-rank test) ($n = 22$). In contrast, treatment of mouse fast-spiking pv cells with ZD7288 ($n = 11$) failed to induce a significant change in Em (median before treatment = −71 mV; IQR, −54 to −74 mV; median after treatment = −72 mV; IQR, −60 to −78 mV; $P = 0.148$ by Wilcoxon signed-rank test), in line with previous studies on rodent neocortical fast-spiking interneurons [42,52,53]. Resting membrane potential data of human and mouse cells can be found in an "Em control and in ZD" worksheet from the S1 Data support file.

In human neurons, the median sag amplitude measured during hyperpolarization from −70 mV to a target potential of −90 mV (median = −90.8 mV; IQR, −87.7 to −92.0 mV) was 3.0 mV (IQR, 1.0 to 5.0 mV; total range, 0 to 15 mV), whereas in mouse basket cells, hyperpolarization from −70 mV to a target potential of −90 mV (median = −90.1 mV; IQR, −89.3 to −91.0 mV) evoked a significantly smaller sag (median = 0.5 mV; IQR, 0 to 1.0 mV; total range, 0 to 3.0 mV; $n = 60$ cells; $P < 0.001$ versus human neurons by Mann–Whitney U test; Fig 1D1). (Sag median amplitude in mouse frontal cortex = 0.65 mV; interquartile range [IQR], 0 to 1.53 mV; $n = 30$. Median sag amplitude in mouse parietal cortex = 0.25 mV; IQR, 0 to 0.75 mV; $n = 30$; $P = 0.135$ using Mann–Whitney U test].

A similar significant difference in sag potential between human and mouse cells was also observed during hyperpolarization from −50 mV to a target of −70 mV (Fig 1D2). In human cells, hyperpolarization from −50 mV (median = −50.3 mV; IQR, −46.9 to −55.0 mV) to −70 mV (median = −70.5 mV; IQR, −68.8 to −72.0 mV) evoked a median negative peak potential of 1.5 mV (IQR, 0.5 to 3.4; total range, 0 to 9.5 mV; $n = 61$ cells), while in mouse cells, a hyperpolarizing command pulse from −50 mV (median = −47.2 mV; IQR, −44.8 to −48.4 mV) to −70 mV (median = −71.0 mV, IQR, −69.2 to −72.5 mV) failed to generate a sag in most

neurons (median = 0 mV; IQR, 0 to 0.2 mV; total range, 0 to 2.1 mV; $P < 0.001$ compared to human cells by Mann–Whitney U test).

To provide further evidence for a physiological function of these HCN channels in regulating electrophysiological responses, we measured somatic HCN channel activity in human and mouse fast-spiking cells ($n = 18$ for both) at the resting membrane potential (Em) (Fig 1D3). Human cells at Em (median = −65.5 mV; IQR, −61.5 to −72.5 mV) were depolarized to a target potential of −50 mV (median = −48.5 mV; IQR, −45.8 to −50.0 mV) to deactivate somatic HCN channels and then repolarized to Em for HCN channel activation [43,44]. This protocol still generated a sag of median amplitude 1.8 mV (IQR, 1.0 to 3.0 mV; total range, 0.9 to 5.1 mV). In mouse basket cells, however, depolarization to a target depolarizing potential (median = −45.8 mV; IQR, −40.1 to −46.6 mV) and repolarization to Em (median = −60.0 mV; IQR, −52 to −69.3 mV) evoked a median sag amplitude of only 0.5 mV (IQR, 0.3 to 1.0; total range, 0 to 1.7 mV; $P < 0.001$ versus human by Mann–Whitney U test). In contrast, the resting membrane potential did not differ significantly between species ($P = 0.09$ using Mann–Whitney U test).

We then examined potential associations between this hyperpolarization-evoked sag and patient clinicodemographic data. There was no significant association between sag amplitude and age from 20 to 82 years (R = 0.139, $P = 0.25$ by Pearson's correlation for sag amplitude versus age, $n = 72$) (Fig 1E1) and no significant trend for specific cortical localization, as a robust sag was found in fast-spiking cells from frontal cortex ($n = 35$), temporal cortex ($n = 24$), and other cortical regions ($n = 13$) including occipital, parietal, and paraventricular areas ($P = 0.13$ using analysis of variance (ANOVA) on ranks) (Fig 1E2). Sag amplitude also did not differ between sexes (female $n = 46$, male $n = 26$, $P = 0.34$, Mann–Whitney U test) or between left and right cerebral hemispheres (left $n = 20$, right $n = 49$, $P = 0.43$ using Mann–Whitney U test). Further, sag amplitudes did not vary according to primary diagnosis for neurosurgery, with robust sag amplitudes in neocortical tissues resected for tumor ($n = 47$), hydrocephalus ($n = 14$), aneurysm ($n = 6$), and other conditions ($n = 5$) ($P = 0.35$ using ANOVA on ranks) (Fig 1E3). Patient data analyses are summarized in Fig 1E1–1E3 and tissue-related clinicodemographic details of cells ($n = 72$) are presented in S1 Table. Collectively, these findings suggest that HCN channel-mediated sag is constitutively and ubiquitously present in the soma of human fast-spiking cortical basket cells at all ages.

## Immunohistochemical evidence for stronger HCN channel localization at the soma membrane of human than mouse fast-spiking pv interneurons

Double immunofluorescent staining for pv, a neurochemical marker specific to fast-spiking interneurons in the neocortex [1,63,64], and HCN1 or HCN2, the two predominant HCN channel isoforms in cortical neurons [21,43,45], suggested that the sag potential measured in human pv cell soma is generated via HCN channels localized at the soma membrane. Indeed, confocal microscopy revealed robust HCN1 and HCN2 immunoreactivity at the human but not mouse pv cell soma membrane (Fig 2A–2D) [65]. We then quantitatively analyzed HCN and cell marker immunoreactivity using confocal images of human and mouse pv cells based on radial immunofluorescence intensity measurement lines (Fig 2A1), where each line measured pv cell-associated fluorophore emission from immunolabeled HCN1 or HCN2 channels over a distance of 8 μm from the cell soma center to the extracellular space (Fig 2A2). In total, 2,340 lines were analyzed from 390 human and mouse pv cells. Human fast-spiking interneurons exhibited intense pv immunofluorescence in the soma, whereas mouse cells exhibited intense somatic emission from tdTomato, the transgenic fluorophore driven by the pv promoter (see Methods). In both cell types, this intracellular emission originated from the nucleus

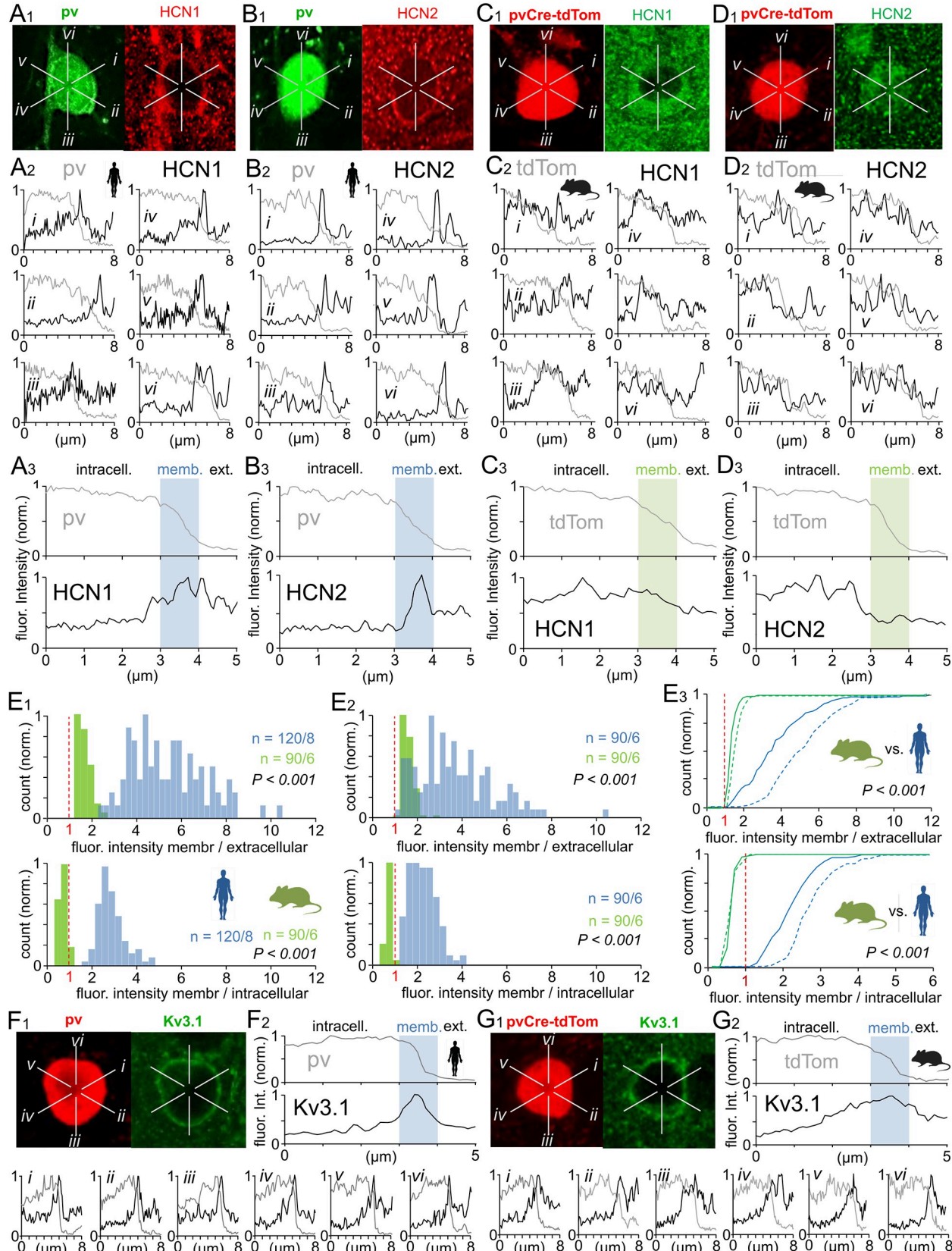

**Fig 2. Immunohistochemical evidence for stronger HCN1 and HCN2 channel localization at somatic cell membrane of human than mouse pv cells.** Confocal microscope study of HCN1 and HCN2 immunofluorescence labeling in somatic region of human and mouse pv cells in L2/3 of neocortex. Panels A-D illustrate similar experiments in human with HCN1 (A1–3) and HCN2 (B1–3), and mouse with HCN1 (C1–3) and HCN2 (D1–3). Immunofluorescence intensity for pv with either HCN1 or HCN2 was measured in a radial pattern of lines diverging from cell soma center and projecting to extracellular space. (**A**) Analysis of pv and HCN1 double-labeling in human cells. (**A1**) Confocal immunofluorescence image of a basket cell in human neocortex L2/3 with pv labeled using Alexa 488 as the fluorophore and HCN1 using cy3 as the fluorophore. Immunofluorescence intensity was measured along six 8-μm long radial lines (*i* to *vi*). (**A2**) Immunofluorescence intensity (ordinate) from pv (gray line) and HCN1 (black line) measured along lines *i* to *vi* starting from the cell soma center (left) and crossing the soma plasma membrane to the extracellular space (right). The pv fluorescence signal is strong within the intracellular space and disappears along measurement lines crossing the plasma membrane zone into the extracellular spaces. HCN1 immunofluorescence, measured in parallel, shows peak intensity in the plasma membrane zone, while pv immunofluorescence simultaneously vanishes in this zone. Ordinates of intensity traces are normalized and scaled similarly to demonstrate the temporal relationship (fluorescence signal peak value = 1). (**A3**) Average pv (top) and HCN1 (bottom) immunofluorescence signals along the six measurement lines for the cell shown in A1–2. Lines are aligned to the midpoint of pv signal descent (see Methods) marking the plasma membrane zone (memb., 1 μm-wide region at pv signal descent illustrated by shaded blue background). (**B**) Analysis of pv and HCN2 double-labeling in human basket cells. (**B1**) Confocal microscope images of pv within the soma labeled by Alexa 488 and HCN2 labeled by cy3. Immunofluorescence measurement lines are labeled *i* to *vi*. (**B2**) Immunofluorescence intensity of pv (gray) and HCN2 (black) measured along lines *i* to *vi* in the cell from B1. Immunoreactivity for HCN2 is strongest along all measurement lines as they cross the extracellular plasma membrane zone, while pv immunofluorescence simultaneously vanishes. (**B3**) Average immunofluorescence signal intensities of pv (top) and HCN2 (bottom) along the measurement lines shown in B1–2. The plasma membrane zone is indicated by a blue background. (**C**) Analysis of the pv-associated intracellular fluorescent signal from tdTomato and from HCN1 immunolabeling in a mouse basket cell. (**C1**) Confocal fluorescence image of a pv-positive (pvCre-tdTom) mouse basket cell immunostained for HCN1 using Alexa 488 as the fluorophore. Radial measurement lines shown as *i* to *vi*. (**C2**) Line analysis of a mouse pv cell expressing tdTomato and also immunostained for HCN1 showing the absence of a clear HCN1 immunofluorescence peak within the plasma membrane zone. (**C3**) Averaged line analyses of pv-associated fluorescent signals (top) and HCN1 immunofluorescence (bottom) for the cell shown in C1-C2. The plasma membrane zone is indicated by green background. (**D**) The pv-associated intracellular fluorescence signal in a mouse cell also immunostained for HCN2. (**D1**) Endogenous intracellular fluorescence driven by the pv promoter (pvCre-tdTomato) and from fluorescence immunostaining for HCN2 (using Alexa 488 as the fluorophore). Measurement lines *i* to *vi* are illustrated on image. (**D2**) Line analyses (*i-vi*) of tdTomato and HCN2 fluorescence signals. This mouse cell emits an intracellular HCN2 signal without a clear signal peak within the apparent extracellular membrane zone. (**D3**) Average of line analyses for D1-D2. Membrane zone shown with green background. (**E**) Histograms summarizing HCN1 and HCN2 immunofluorescence intensity levels at the soma plasma membrane for 390 pv cells (analyzed like the sample cells in A-D). Bin size is 0.25. (**E1**) Top: HCN1 immunofluorescence intensity at the plasma membrane zone versus the extracellular space for human pv-positive basket cells (blue bars) and mouse pv-expressing basket cells (green). Both human and mouse cells show higher fluorescence intensity at the membrane zone than the extracellular space, but human basket cells also show a significantly higher ratio of membrane-bound signal compared to mouse basket cells ($P < 0.001$ between species). Bottom: HCN1 immunofluorescence intensity at the soma plasma membrane compared to the intracellular space. Human cells show higher HCN1 immunofluorescence at the plasma membrane than in the intracellular space (ratio > 1), whereas mouse cells exhibit stronger HCN1 immunofluorescence within the intracellular space (cytoplasm) than at the membrane (ratio < 1) ($P < 0.001$ for the ratio between species by Mann–Whitney U test). Ratio of 1 at abscissa is indicated in red and with vertical dotted line. The *n* values indicate the numbers of analyzed cells, patients, or mice. (**E2**) Top: The HCN2 immunofluorescence intensity at the cell membrane zone versus extracellular space for human pv-positive basket cells (blue bars) and mouse pv-expressing basket cells (green). Both human and mouse pv cells show stronger HCN2 signals at the cell membrane than in the extracellular space, but human cells show a higher ratio of membrane-bound signal ($P < 0.001$ ratio between species). Bottom: HCN2 immunofluorescence intensity at the cell membrane versus intracellular space. Human cells show stronger signals at the cell membrane than inside the cell. Mouse cells show stronger HCN2 immunofluorescence in the intracellular space than at the cell membrane (ratio < 1) ($P < 0.001$ ratio between species by Mann–Whitney U test). Ratio of 1 at abscissa is indicated in red and with vertical dotted line. (**E3**) Comparison of HCN1 and HCN2 immunofluorescence localization results for pv cells between species. Top: Cumulative histogram summarizing the membrane versus extracellular site measurements in human (blue line) and mouse (green line) for HCN1 (solid line) and HCN2 (dotted line) ($P < 0.001$ between species). Bottom: HCN1 and HCN2 immunofluorescence intensity at the cell membrane versus intracellular space ($P < 0.001$ between species). (ANOVA on ranks with post hoc Dunn's pairwise test). The data shown in Fig 2E1–2E3 histograms are available in S1 Data. (**F-G**) Verification of the plasma membrane zone in human (F) and mouse (G) images by immunofluorescence labeling for Kv3.1 potassium channels, which are characteristically enriched at the somatic membrane. (**F1**) Top: Confocal immunofluorescence image of pv immunoreactivity in a human L2/3 cell (cy3 as the fluorophore) costained for Kv3.1 (Alexa 488 as the fluorophore). Immunofluorescence intensity was measured along radial lines *i-vi*. Bottom: Line analysis of lines *i-vi* for pv and Kv3.1 immunofluorescence. Kv3.1 signal shows peak intensity at the plasma membrane zone, whereas pv immunofluorescence disappears within this zone. (Intensity traces normalized to the fluorescence signal peak value.) (**F2**) Average of line analyses from F1. (**G1**) Top: Confocal fluorescence image of a mouse basket cell showing fluorescence signals from pv (pvCre-tdTomato) and Kv3.1 (Alexa Fluor 488). Also shown are radially patterned intensity measurement lines *i-vi*. Bottom: Line analyses (*i-vi*) and (G2) average of aligned fluorescence intensity traces showing the Kv3.1 signal peak at the plasma membrane zone.

and cytoplasm. In line analysis, the plasma membrane area appeared as a narrow zone where pv (or tdTomato) fluorescence signal rapidly disappeared (Fig 2A2–2A3). The immunofluorescence intensities of labeled HCN1 and HCN2 channels were measured along with the intracellular marker signal and were quantified in the cytoplasm, at the somatic membrane zone (defined as a 1-μm wide region where intracellular signal disappeared; see Methods for details), and in the extracellular space (defined by the absence of intracellular signal), as illustrated in

Fig 2A3. Radial pattern line analysis of HCN1 and HCN2 expression by human pv cells is shown in Fig 2A1–2A3 and 2B1–2B3, whereas mouse results are presented in Fig 2C1–2C3 and 2D1–2D3.

Human pv cell HCN1 immunofluorescence intensity (in arbitrary units) was 3.42-fold (IQR, 2.56- to 4.78-fold) higher at the cell soma membrane zone than that in the extracellular space and 2.25-fold (IQR, 1.77- to 2.65-fold) higher than that in the cell soma intracellular space ($n = 720$ lines in 120 cells sampled from 8 patient tissue samples, 15 cells per sample) (Fig 2E1). Similarly, HCN2 immunofluorescence intensity was 5.22-fold (IQR, 4.01- to 6.56-fold) higher at the membrane zone than that in the extracellular space and 2.74-fold (IQR, 2.42- to 3.18-fold) higher than that in the soma intracellular space ($n = 540$ lines in 90 cells sampled from 6 patient tissue samples, 15 cells per sample) (Fig 2E2). Mouse pv cells also presented stronger HCN1 and HCN2 immunofluorescence signals at the cell body plasma membrane than that in the extracellular space (HCN1 median = 1.51-fold; IQR, 1.41- to 1.68-fold; HCN2 median = 1.58-fold; IQR, 1.43- to 1.83-fold; $n = 540$ and 540 lines in 90 cells sampled from 6 mouse tissue samples, 15 cells per sample). However, intensities in mouse soma membrane (compared to extracellular) were significantly weaker than those in human soma membrane zone ($n = 90$ and 120, $P < 0.001$ and $< 0.001$, respectively, using Mann–Whitney U test) (Fig 2E1–2E2). Unlike human pv cells, mouse pv cells exhibited stronger HCN1 and HCN2 signals in the somatic intracellular space (see Fig 2C3 and 2D3), with the plasma membrane intensity of HCN1 and HCN2 emission reaching 0.70-fold (IQR, 0.65- to 0.75-fold) and 0.63-fold (IQR, 0.56- to 0.73-fold), respectively, of that of the intracellular somatic space (Fig 2E1–2E2). Fig 2E3 presents statistical comparisons of HCN1 and HCN2 immunofluorescence intensities between species for membrane versus extracellular space measurements ($P < 0.001$ between species using ANOVA on ranks with Dunn's post hoc pair-wise test) and membrane versus intracellular space measurements ($P < 0.001$ between species by the same tests). We also validated the cell membrane zone definition used for HCN1 and HCN2 immunofluorescence intensity line analysis by evaluating the localization of Kv3.1 potassium channels, which are also known to be enriched at the somatic membrane of fast-spiking pv neurons [52,66,67]. Fig 2F and 2G show immunoreactivity patterns for Kv3.1 based on fluorescence intensity line analysis of sample human and mouse pv cells, respectively. In both human and mouse pv cells, Kv3.1 immunofluorescence was localized to a narrow zone at the edge of the intracellular pv fluorescence signal. The clinicodemographic data of patients providing these tissue samples for immunofluorescence studies are summarized in S1 Table.

To provide additional evidence that HCN immunofluorescent signals arise from the membrane surface, immunofluorescence localization was investigated in selected human and mouse tissue sections using appropriate antibodies and the dSTORM super-resolution technique (see Methods) [68], which can achieve a spatial resolution of <0.1 μm for fluorophore signal analysis. These dSTORM immunofluorescence images showed antibody binding (fluorescence signals) concentrated in a narrow layer around the human pv neuron cell body and clear contrast between apparent cell membrane areas and both extra- and intracellular spaces for HCN1 (Fig 3A1–3A2), HCN2 (Fig 3B1–3B2), and Kv3.1 (Fig 3C1–3C2). However, similar images of mouse pv cells showed punctate regions of HCN1 and HCN2 fluorophore signal spread across the entire cell body rather than at the soma membrane, likely reflecting clusters within the cytoplasm [65]. Fig 3D and 3E shows dSTORM images of typical mouse pv cells lacking clear immunofluorescence signal contrast between apparent membrane areas and both extra- and intracellular spaces for HCN1 (Fig 3D1–3D2) and HCN2 (Fig 3E1–3E2). Conversely, dSTORM imaging of Kv3.1 immunoreactivity in mouse tissue revealed fluorophore molecules concentrated in a narrow layer around the pv cell body in apparent extracellular membrane spaces (Fig 3F1–3F2), similar to Kv3.1 immunoreactivity in human pv cells.

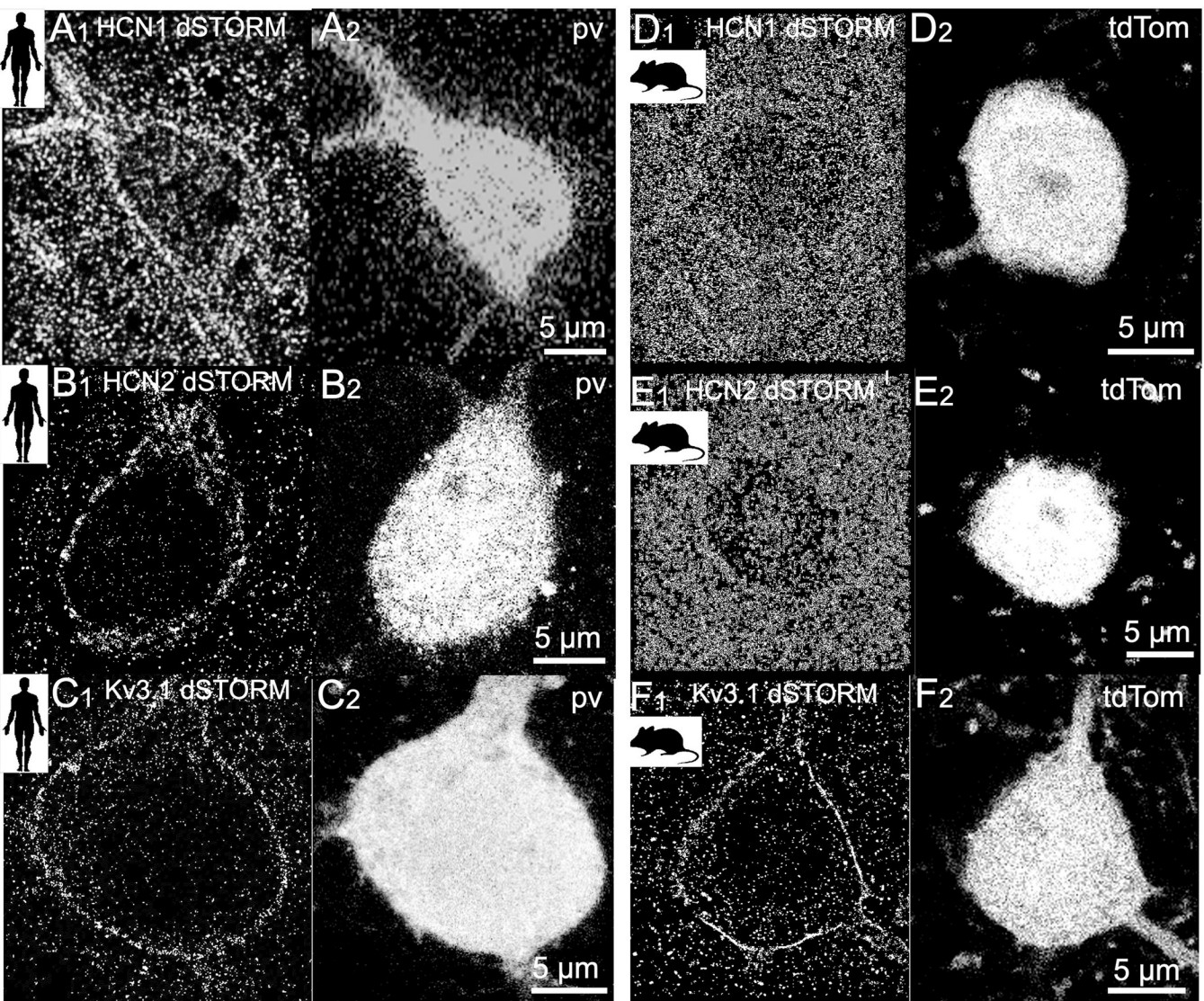

**Fig 3. Visualization of somatic HCN1, HCN2, and Kv3.1 channels in human and mouse pv-immunopositive interneurons by dSTORM super-resolution immunofluorescence microscopy.** (**A-C**) dSTORM super-resolution images showing high-resolution immunofluorescence labeling of HCN1, HCN2, and Kv3.1 channels in human pv-immunopositive cells. Pv immunoreactivity is shown in paired confocal images. (**A**) dSTORM image of HCN1 immunolabeling (**A1**) using Alexa 647 as the fluorophore paired with a confocal image of pv immunoreactivity (**A2**) using Cy3 as the fluorophore. HCN1 immunoreactivity is concentrated at apparent membrane zone around the cell soma and proximal dendrites. (**B**) dSTORM image of HCN2 immunolabeling (**B1**) using Alexa 647 as the fluorophore paired with a confocal image of pv immunoreactivity (**B2**) using Cy3 as the fluorophore. Strongest HCN2 immunofluorescence is localized at the apparent cell soma membrane zone. (**C**) dSTORM image of Kv3.1 potassium channel immunostaining using CF568 as the fluorophore (**C1**) paired with a confocal image of pv immunoreactivity (**C2**) using Alexa 488 as the fluorophore. Kv3.1 immunoreactivity is concentrated at the apparent membrane zone. (**D-E**) Mouse pv cell showing weak or absent HCN1 or HCN2 immunofluorescence at the cell soma membrane. dSTORM images of HCN1 (**D1**) and HCN2 (**E1**) immunoreactivity using Alexa 647 as the fluorophore are paired with confocal images of pv-associated fluorescence (tdTomato) (D2 and E2, respectively). (**F**) dSTORM image of Kv3.1 immunolabeling (**F1**) using Alexa 647 as the fluorophore paired with a confocal image of pv-associated intracellular fluorescence signal (**F2**). Kv3.1 immunofluorescence is localized around the cell soma.

## Electrophysiological evidence for more robust HCN channel activity at the somatic cell membrane of human than mouse fast-spiking interneurons

Direct evidence for distinct HCN channel distributions on the soma membrane of human fast-spiking pv neurons compared with those of mouse pv neurons was obtained via outside-

out patch recordings. We initially obtained recordings from 13 human fast-spiking cells in whole-cell current-clamp mode to measure sag potential and then gently retracted the micro-pipette under visual guidance from an online camera image until the cell soma with nucleus and cytoplasm inside was freed from the tissue (in 1 of the 13 experiments, we failed to entrap the nucleus inside the patch). In this outside-out patch configuration, every examined cell soma stopped firing full amplitude action potentials in response to depolarizing current steps (Fig 4A1) and input resistance increased from 103.0 MΩ (IQR, 69.7 to 145.2 MΩ) in whole-cell mode to 241.3 MΩ (IQR, 98.3 to 397.8 MΩ) ($P < 0.001$ using Wilcoxon signed-rank test) (Fig 4A2–4A3). However, human neurons exhibited a median sag amplitude of 3.86 mV (3.01 to 5.87 mV) in the whole-cell mode and 2.64 mV (2.28 to 3.31 mV) in the outside-out patch configuration during hyperpolarizing steps to −90 mV from −60 mV ($n = 13$) ($P < 0.001$ using Wilcoxon signed-rank test) (Fig 4B1–4B2). Although the relative increase in input resistance (outside-out versus whole-cell) due to outside-out patch formation varied among cells, the change did not correlate with the sag amplitude change (R = −0.178, $P = 0.616$, Spearman's rank order correlation, $n = 13$ cells), indicating that outside-out patches of different relative size (patched membrane area) still consistently retained the sag (Fig 4B3). In these experiments, neurons were identified using their electrophysiological fast-spiking characteristics because visualization was unsuccessful after the pull-out procedure. S1 Table lists the fast-spiking features of individual cells in outside-out patch experiments and corresponding patient data.

We performed similar experiments in mouse pv cells (Fig 4C1). Similar to human cells, input resistance increased from 59.0 MΩ (IQR, 55.0 to 71.5 MΩ) in the whole-cell mode to 124 MΩ (IQR, 86.3 to 153.5 MΩ) in outside-out patches ($P = 0.008$ using Wilcoxon signed-rank test) (Fig 4C2). This increase was associated with a moderate reduction in the already modest sag amplitude from 0.80 mV (IQR, 0.75 to 1.30 mV) to 0.65 (IQR, 0.53 to 0.80 mV) ($P = 0.0016$, $n = 8$ using Wilcoxon signed-rank test) (Fig 4C3). Direct comparisons revealed that the sag amplitude measured in somatic outside-out patches in human cells was significantly greater than that in mouse fast-spiking cells ($P < 0.001$ using Mann–Whitney U test), although the relative reduction was similar after outside-out patch formation (human median = 70.6%, IQR, 60.9% to 81.9%; mouse median = 70.8%, IQR, 54.1% to 91.4%; $P = 0.971$ using Mann–Whitney U test). These results indicate that HCN channel density is higher at the soma membrane of human fast-spiking cells than that of mouse fast-spiking cells; however, the distribution pattern is similar.

To further demonstrate that HCN channels are active at or in close proximity to the cell soma and substantially modulate somatic electrophysiological properties, we examined the contributions of these channels to the somatic leakage conductance (Gleak) as this property influences resting membrane potential, input resistance, and membrane time constant [43]. The HCN blocker ZD7288 (10 μM) significantly decreased Gleak in the soma of 4 of 5 human cells when voltage-clamped at the resting potential (mean Em = −59.0 mV) but in only 1 of 5 mouse cells (mean Em = −70.6 mV) ($P < 0.05$, Wilcoxon signed-rank test) as measured by brief voltage-clamp steps (−10 mV, 10 ms applied at 0.1 Hz). Fig 4E and 4F describe the leak conductance analysis and illustrate sample experiments in tissue samples from human (Fig 4E1) and mouse (Fig 4F1). The cells were voltage-clamped at their baseline Em value also in the presence of ZD. In the four human basket cells with significant effects of ZD7288 on Gleak, the leak conductance decreased from a median of 4.08 nS (IQR, 3.42 to 4.50 nS) to 3.53 nS (IQR, 2.25 to 3.96 nS) during the 5-min drug wash-in period, corresponding to a median 14.9% decrease (IQR, 8.9% to 36.6%), while one cell failed to show Gleak change (Fig 4E2–4E3). In mouse cells, Gleak was 4.87 nS (4.75 to 6.17 nS) at baseline ($n = 5$) and decreased significantly during ZD7288 wash-in in only one cell (by 7.6%; $P = 0.008$ by Wilcoxon signed-rank test)

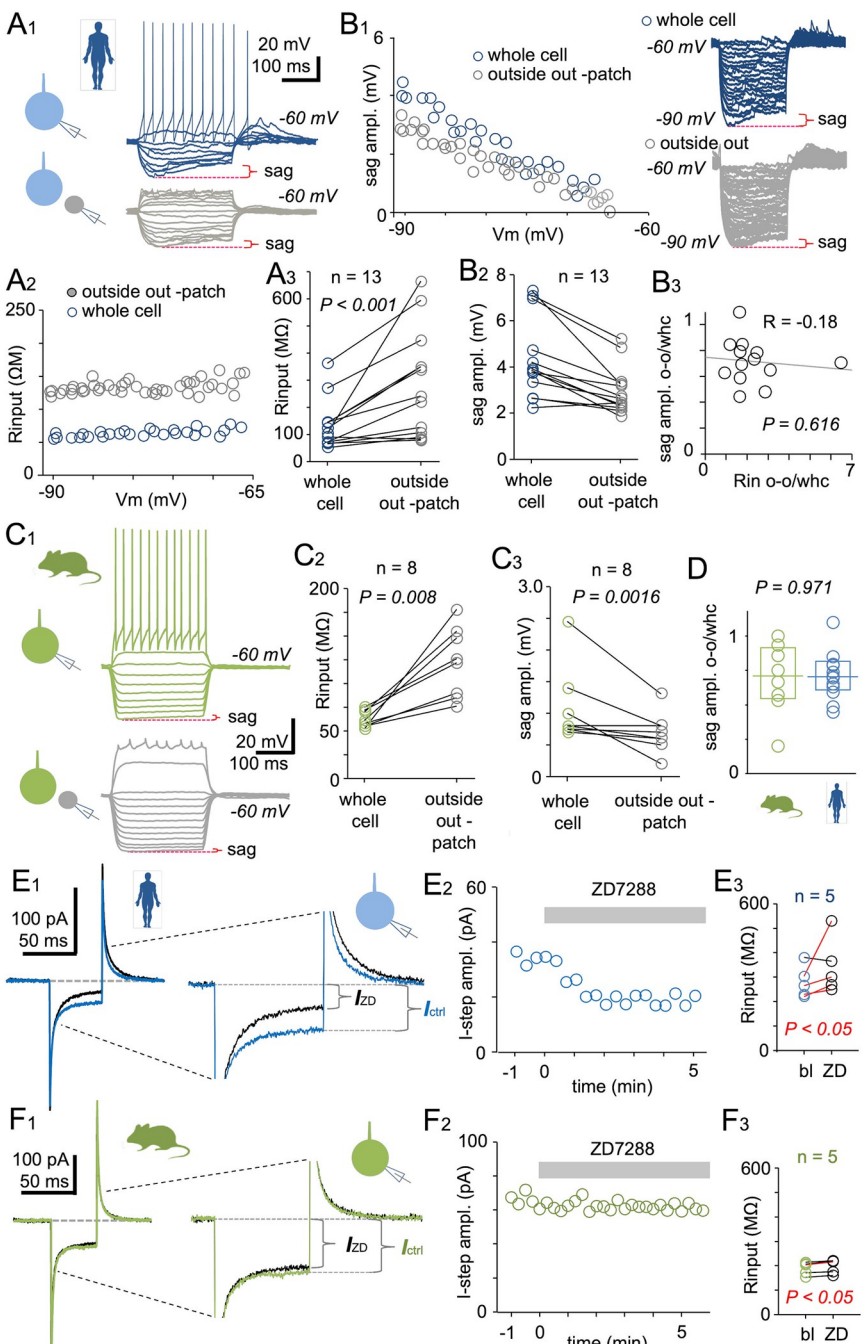

**Fig 4. Evidence for stronger HCN channel activity at the somatic compartment of human fast-spiking interneurons compared to mouse fast-spiking interneurons from outside-out patch recordings. (A)** Outside-out patch recordings obtained from human fast-spiking cell somata. **(A1)** First, the sag potential was measured in somatic whole-cell clamp mode (blue traces) by applying hyperpolarizing current steps from −60 mV. Sag potential amplitude during a voltage step to −90 mV is indicated by the red marking. Next, outside-out nucleated patches were established by pulling out the pipette. Gray traces show membrane potential changes from current steps applied to the same cell but in the somatic outside-out patch configuration. Schematic insets on left show the experimental design. **(A2)** Input resistance (ordinate) of the same cell measured by membrane potential steps to hyperpolarized potentials (abscissa) from −60 mV in the somatic whole-cell clamp mode (blue symbols) and outside-out patch configuration (gray symbols). **(A3)** Input resistance in 13 fast-spiking human basket cells measured by hyperpolarizing step to −90 mV from −60 mV in whole-cell clamp mode (blue symbols) and the outside-out patch configuration (gray symbols) ($P < 0.001$ by Wilcoxon signed-rank test). **(B)** Sag amplitude is largely preserved in human fast-spiking cell somatic outside-out recordings. **(B1)** Sag amplitude (ordinate) in one fast-spiking cell measured during membrane potential

steps to hyperpolarizing potentials (ordinate) from −60 mV in somatic whole-cell recording mode (blue symbols) and the outside-out patch configuration (gray symbols). Left: Plot shows incremental increases in sag amplitude at more hyperpolarized membrane potentials in both whole-cell mode (blue) and the outside-out patch configuration (gray). Right: Traces show hyperpolarizing membrane potential steps from −60 mV in the two modes. Sag amplitude during steps to −90 mV is indicated by red marking. (**B2**) Sag amplitude measured during hyperpolarizing steps to −90 mV from 13 fast-spiking human cells in whole-cell clamp mode (blue) and the outside-out patch configuration (gray). Sag amplitude is only moderately reduced in outside-out patches ($P < 0.001$ by Wilcoxon signed-rank test). (**B3**) Sag amplitude change (ordinate) does not correlate with the relative increase in soma input resistance (Rin, abscissa) due to the change from somatic whole-cell clamp (whc) to outside-out patch (o-o) (R = −0.178, $P = 0.616$, Spearman's rank order correlation). (**C**) Outside-out patch recordings from mouse fast-spiking pv cell somata. (**C1**) Small somatic sag potential measured in whole-cell clamp (green traces) in response to hyperpolarizing current steps from −60 mV. Sag potential amplitude during voltage step to −90 mV is indicated by red marking. Gray traces show membrane potential changes from current steps applied to the same cell after establishing the somatic outside-out patch configuration. Schematic shows the experimental design. (**C2**) Input resistance (Rinput) of 8 fast-spiking mouse cells measured by hyperpolarizing step to −90 mV from −60 mV in whole-cell clamp mode (green symbols) and the outside-out patch configuration (gray symbols) ($P = 0.008$ by Wilcoxon signed-rank test). (**C3**) Sag amplitude (measured during steps to −90 mV) of 8 fast-spiking mouse pv-expressing fast-spiking cells in whole-cell clamp mode (green) and outside-out patch configuration (gray). Sag amplitude is preserved although reduced in outside-out patches ($P < 0.0016$ by Wilcoxon signed-rank test). (**D**) On average, the sag amplitude was similarly preserved in outside-out patches compared to the preceding whole-cell recording in both mouse (green symbols) and human cells (blue symbols) ($P = 0.971$ by Mann–Whitney U test). Box plots show median and upper and lower quartiles. (**E-F**) Voltage-clamp recording at the cell soma showing a greater reduction of leak conductance (Gleak) in human than mouse fast-spiking basket cells during application of the HCN channel blocker ZD7288 (ZD, 10 μM). Cells were voltage-clamped at the resting membrane potential of baseline control condition throughout experiment (average of −59.0 mV for human and −70.6 mV for mouse cells). (**E1**) Sample traces (averages of five) showing somatic whole-cell voltage-clamp current evoked by −10 mV steps (10 ms in duration) from Em in a human fast-spiking neuron. Blue trace is the response under control conditions and black trace is that in the presence of ZD7288 (10 μM). Voltage-clamping of a fast-spiking basket cell at −65 mV. Trace holding-current levels in the figure are aligned for comparison of the step amplitude change induced by ZD. Zoomed image illustrates the current response to a −10-mV voltage-clamp step under control conditions and in the presence of ZD. The current amplitude (shown as $I_{ctrl}$) is smaller during the same voltage step in the presence of ZD (shown as $I_{ZD}$) as indicated by brackets. $I_{ctrl}$ or $I_{ZD}$ amplitude is proportional to somatic leak conductance. Inset shows a schematic of the clamp protocol in human cell. (**E2**) Plot showing the current amplitudes evoked by −10 mV voltage steps before and during ZD wash-in (indicated with gray bar). (**E3**) Somatic input resistance, measured as 1/Gleak, in five human fast-spiking cells at baseline and following ZD7288 wash-in. In four of five cells (red lines), the change is significant ($P < 0.05$ by Wilcoxon signed-rank test). One cell without significant change is indicated by black line. Values are averages from 1-min time windows before ZD application (baseline, bl) and after 5 min in the presence of ZD (ZD). (**F**) Mouse fast-spiking cells regularly fail to show a change in leak conductance at the cell soma under HCN channel blockade. (**F1**) Sample traces (average of five) showing whole-cell voltage-clamp current responses to −10 mV voltage steps (10 ms) from Em in a mouse fast-spiking pv neuron. Green trace is under baseline control conditions, and black trace is in the presence of ZD7288 (10 μM) at −69 mV. Zoomed image shows the current responses to a −10-mV step at baseline and in the presence of ZD. There is a negligible change in current amplitude following ZD wash-in. The schematic inset indicates whole-cell recording from mouse cell. (**F2**) Somatic current amplitudes in response to −10 mV voltage steps before and during ZD wash-in (indicated by gray bar). (**F3**) Somatic input resistance (1/Gleak) in five mouse fast-spiking pv cells at baseline (bl) and following ZD7288 wash-in (ZD) (averages from 1-min time windows before ZD application and after 5 min in the presence of ZD). Input resistance is significantly changed in only one cell (red line) of five ($P < 0.05$ by Wilcoxon signed-rank test). The data shown in Fig 4E and 4F can be found in S1 Data.

(Fig 4F2–4F3). Fig 4E3 and 4F3 summarize this by showing cell input resistance values (as 1/Gleak, measured in voltage-clamp for −10 mV, 10 ms step) at baseline (human median = 265 MΩ; IQR, 224 to 342 MΩ. Mouse median = 205 MΩ; IQR, 163 to 211 MΩ) and following wash-in of ZD7288 (human median = 301 MΩ; IQR, 258 to 447 MΩ. Mouse median = 217 MΩ; IQR, 168 to 220 MΩ) in the five examined human and mouse cells each.

## HCN channels shorten the spike generation lag by accelerating somatic membrane potential kinetics in human fast-spiking neurons

Next, we studied the effect of HCN channel blockade on somatic membrane potential kinetics and measured the apparent membrane time constant (τ) in human basket cells. The median τ as measured by applying hyperpolarizing steps from −70 mV to −90 mV (Fig 5A1) was 7.92 ms (IQR, 5.88 to 10.94 ms) at baseline and 11.38 ms (IQR, 7.20 to 15.44 ms) following 5-min

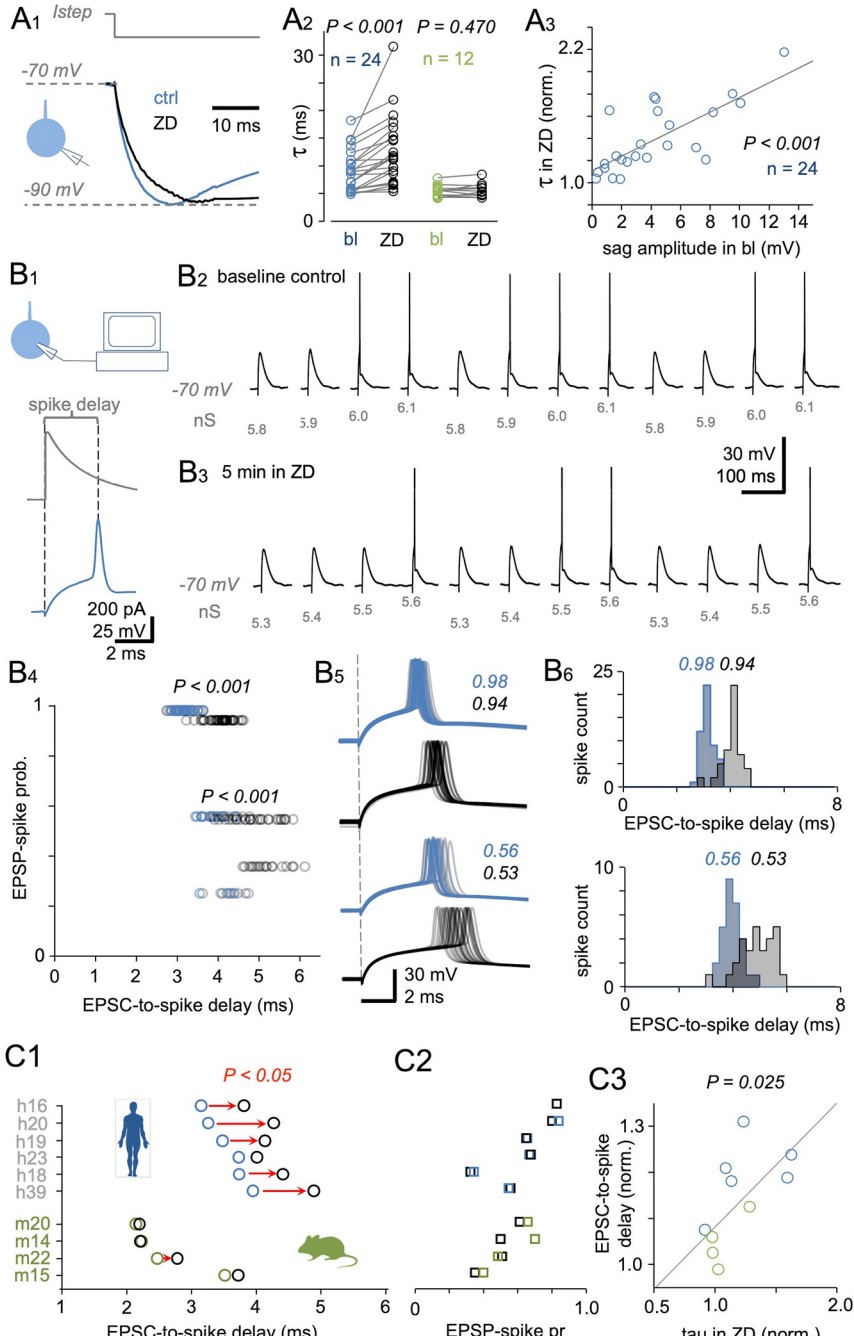

**Fig 5. HCN channel blockade lengthens the somatic time constant and somatic spike generation delay in human fast-spiking interneurons.** (**A**) The HCN channel blocker ZD7288 slows the time course of membrane potential changes in human fast-spiking interneuron soma. (**A1**) Somatic membrane apparent time constant (τ) measured from equal-amplitude hyperpolarizing membrane potential steps from −70 mV to −90 mV in whole-cell clamp mode (illustrated in inset schematic) under control conditions (blue, average of five traces from one cell) and in the presence of ZD7288 (black). *Istep* shows time course of the hyperpolarizing square-pulse current step. (**A2**) Plot summarizing τ as measured using standard hyperpolarizing membrane potential steps for human and mouse fast-spiking cells (A1) at baseline control (bl) and following ZD7288 (ZD) infusion. τ is significantly lengthened by ZD in human (blue, $P < 0.001$, $n = 24$ cells) but not mouse (green) fast-spiking cells ($n = 12$, $P = 0.470$ by Wilcoxon signed-rank test). (**A3**) The effect of ZD7288 on somatic membrane τ is strongest in cells showing the largest HCN sag potential amplitude at baseline. Plot shows the baseline-normalized membrane τ value in the presence of ZD (ordinate) against the sag amplitude (abscissa) measured before ZD wash-in. The two parameters show a strong correlation ($n = 24$, R = 0.73, $P < 0.001$ by Spearman's rank order correlation analysis). Gray line shows regression. (**B**) ZD prolongs the somatic

action potential generation delay in human fast-spiking cells. (**B1**) Excitatory postsynaptic currents (EPSCs) resembling natural EPSCs recorded from the somata of human fast-spiking interneurons were generated at the soma by dynamic clamp to elicit EPSPs reaching firing threshold. Traces illustrate EPSPs with spikes (blue) elicited by EPSCs (gray, dynamic clamp) at −70 mV. Dotted vertical lines and the bracket show EPSC-to-spike lag (from EPSC onset to spike peak). Schematic inset illustrates the experimental setup for dynamic clamp. (**B2**) Dynamic clamp parameters were set to generate incrementally greater EPSC amplitudes to evoke EPSPs with spike probabilities <1. Traces show EPSPs starting from subthreshold and reaching beyond the firing threshold by applying dynamic clamp input in increments of 0.1 nS from 5.8 nS to 6.1 nS (dynamic clamp input strengths indicated below EPSPs) at an interval of 4 s. Traces illustrate three consecutive cycles for one cell at baseline with EPSP–spike probability <1 in response to the second weakest dynamic clamp input (5.9 nS). Resting membrane potential in the experiment was kept at −70 mV. The cycle of four incremental EPSPs was repeated 30 times. (**B3**) Recording from the same cell following 5 min ZD wash-in (resting membrane potential kept at −70 mV). Cell input resistance increased, so EPSC strengths were readjusted to elicit an EPSP–spike probability <1. Traces show three consecutive cycles of incrementally increasing EPSP amplitude with EPSP–spike probability <1 in response to 5.5 nS input to dynamic clamp. (**B4**) EPSP–spike probability plotted against the spike delay measured during 30 cycles in one fast-spiking human basket cell. Plot shows spikes generated by three EPSP strengths with spike probability <1 at baseline (blue dots) and following 5 min ZD wash-in (black symbols). *P* values show the difference between spike time delay values evoked at a similar spike probability under baseline conditions and in the presence of ZD. (**B5**) Superimposed EPSP–spike traces illustrated in the same cell at baseline (blue) and in the presence of ZD (black). EPSP–spike probabilities shown in the inset. (**B6**) Histograms summarizing EPSC-to-spike delay (from EPSC onset to spike peak) values for the spikes in B5. (**C1**) Plot showing averaged EPSC-to-spike responses in six human cells (blue) and four mouse cells (green) at baseline and following ZD wash-in (black). Identifying experimental codes are shown on the ordinate (see S1 Table). Red arrows indicate direction of a significant change (*P* < 0.05 by Wilcoxon signed-rank test). (**C2**) The EPSP–spike probabilities for the EPSC-to-spike delay data in C1 at baseline (blue, green) and in the presence of ZD (black). (**C3**) The effect of ZD on spike lag is largest in cells showing the greatest increase in membrane time constant under ZD treatment (R = 0.685 *P* = 0.025 by Spearman's rank order correlation, *n* = 10 cells). Blue, human cells; green, mouse cells. Gray line indicates the regression. The underlying data supporting Fig 5B4'5B6 and 5C1–2, and data illustrated in Fig 5A– 5C can be found in S1 Data.

wash-in of ZD7288 (*P* < 0.001 versus baseline by Wilcoxon signed-rank test, *n* = 24), showing a 1.44-fold increase (Fig 5A2). Notably, this lengthening of τ by ZD7288 was significantly correlated with sag potential amplitude measured at baseline (R = 0.73, *P* < 0.001 by Spearman's rank order correlation, *n* = 24 cells) (Fig 5A3). In comparison, treatment of mouse fast-spiking pv cells (*n* = 12) with ZD7288 failed to have a significant effect, as τ was 5.76 ms (IQR, 4.88 to 6.05 ms) at baseline and 5.62 ms in the presence of the drug (IQR, 5.05 to 6.50 ms) (*P* = 0.470 by Wilcoxon signed-rank test).

Treatment with ZD7288 also significantly increased the somatic input resistance of human fast-spiking basket cells from 163 MΩ (IQR, 142 to 197 MΩ) to 226 MΩ (IQR, 167 to 309 MΩ) as measured by the current pulse amplitude generating a voltage step to −90 mV from −70 mV (*P* < 0.001, Wilcoxon signed-rank test, *n* = 24 cells). In contrast, the input resistance of mouse fast-spiking neurons was not changed significantly by ZD7288 treatment (147 MΩ, IQR, 106 to 176 MΩ versus 161 MΩ, IQR, 120 to 185 MΩ; *P* = 0.176 by Wilcoxon signed-rank test, *n* = 12). Thus, despite no difference at baseline between human and mouse basket cells (*P* = 0.120 by Mann–Whitney U test), somatic input resistance was significantly higher in human cells under HCN blockade (*P* = 0.011 by Mann–Whitney U test). This finding is consistent with lower baseline Gleak under HCN blockade [20,22,35,69] and further suggests that somatic HCN channels are an important target for regulation of temporal summation and EPSP–spike transformation kinetics in the human pv cell soma.

To directly examine possible effects of HCN channel blockade on the kinetics of somatic input–output transformation, we measured the delay between EPSP and somatic action potential generation under control conditions and following ZD7288 application using a dynamic clamp protocol [19,70]. To measure effects mediated primarily by the change in membrane potential kinetics from HCN blockade (as observed by changes in τ), the cell membrane potential was clamped at −70 mV both at baseline and in the presence of ZD7288 to eliminate the influence of ZD7288-induced hyperpolarization; moreover, EPSP-evoked spike probability

was kept constant under both conditions by readjusting excitatory synaptic current (EPSC) strength in the presence of ZD7288 to compensate for the input resistance change. Fig 5B1 illustrates how conductance was applied to dynamic clamp input to generate excitatory currents in cell soma mimicking the glutamatergic EPSCs reported in human fast-spiking cell soma by whole-cell recordings that generate spikes [11,13,18,19,25]. Then, incremental amplitude EPSPs were generated until action potential firing, and four dynamic clamp input conductance strengths were set to evoke EPSPs and firing with probability <1 (Fig 5B2). The induction of incrementally larger amplitude EPSPs was repeated at least 30 times at baseline and then repeated following 5 min of ZD7288 wash-in and readjustment to evoke EPSPs with firing probabilities <1 (Fig 5B3). Analysis of spike probability and EPSC-to-spike delay (defined as the delay from EPSC onset to spike peak) under baseline and drug wash-in conditions (Fig 5B4) showed that the delay was significantly increased by ZD7288 (Fig 5B5-6) in five out of six human basket cells ($P < 0.005$ by Wilcoxon signed-rank test) but in only one of four mouse basket cells (Fig 5C1).

As illustrated in Fig 5C1, ZD7288 prolonged the EPSC-to-spike delay of the 5 responsive human cells from 3.48 ms (IQR, 3.20 to 3.81 ms) to 4.27 ms (IQR, 3.97 to 4.65 ms) (median spike delay was also numerically increased in the sixth cell from 3.73 ms to 4.01 ms), for a mean 23% increase. Note that these delay data were collected from EPSPs at baseline and ZD7288 cycles showing similar EPSP–spike probability (Fig 5C2). In similar experiments on mouse cells, only one of four mouse cells demonstrated a significant increase in delay following ZD7288 treatment (from 2.47 ms to 2.80 ms, $P < 0.05$ by Wilcoxon signed-rank test) (Fig 5C1). The spike delay results in all human ($n = 6$) and mouse cells ($n = 4$) are shown in a "Metadata for 5C1" worksheet from the S1 Data support file. Notably, we found a strong correlation between ZD7288-evoked lengthening of the spike delay and its effect on the somatic membrane time constant (measured as shown in Fig 5A1), suggesting that the shorter spike delay at baseline in response to somatic EPSCs results from faster membrane kinetics due to HCN activity at a fixed membrane potential of −70 mV (R = 0.685 $P$ = 0.025, Spearman's rank order correlation, $n$ = 6 human and 4 mouse cells) (Fig 5C3).

## A model neuron with experimentally derived electrophysiological parameters also demonstrates accelerated input–output function from simulated perisomatic HCN activity

To provide theoretical support for the functional importance of HCN channel activity in human cortical basket cells, we constructed a model neuron with a somatic compartment, primary dendrite, and axon [71]. The axonal and dendritic cable compartments were designed with a length of 200 μm and diameter of 1.2 μm. The model cell included a persistent noninactivating leak current in all compartments as well as standard Hodgkin–Huxley-type spike-generating $Na^+$ and $K^+$ conductances and a slowly activating M-type $K^+$ current in axonal and somatic compartments. The M-type $K^+$ current reproduced the gradual increase in interspike interval observed in biological neurons during positive current steps up to +40 pA (Fig 6A1 and 6B1). Somatic and dendritic compartments also contained HCN current and inward-rectifying $K^+$ current (Kir) [72,73]. The Kir conductance replicated the decreased input resistance observed during the application of hyperpolarizing currents from −20 to −100 pA in the presence of ZD7288 (Fig 6A2 and 6B2). We assumed that 65% of the sag amplitude in the soma was mediated by local HCN channels, as indicated via our outside-out experiments (see Fig 4), whereas the remaining 35% was generated in the model by conductance of HCN channels (gHCN) localized at the primary dendrite. We set model parameters based on the real responses of sample cell h35 (Fig 6A1–6A3), a fast-spiking basket cell showing a 5-mV sag

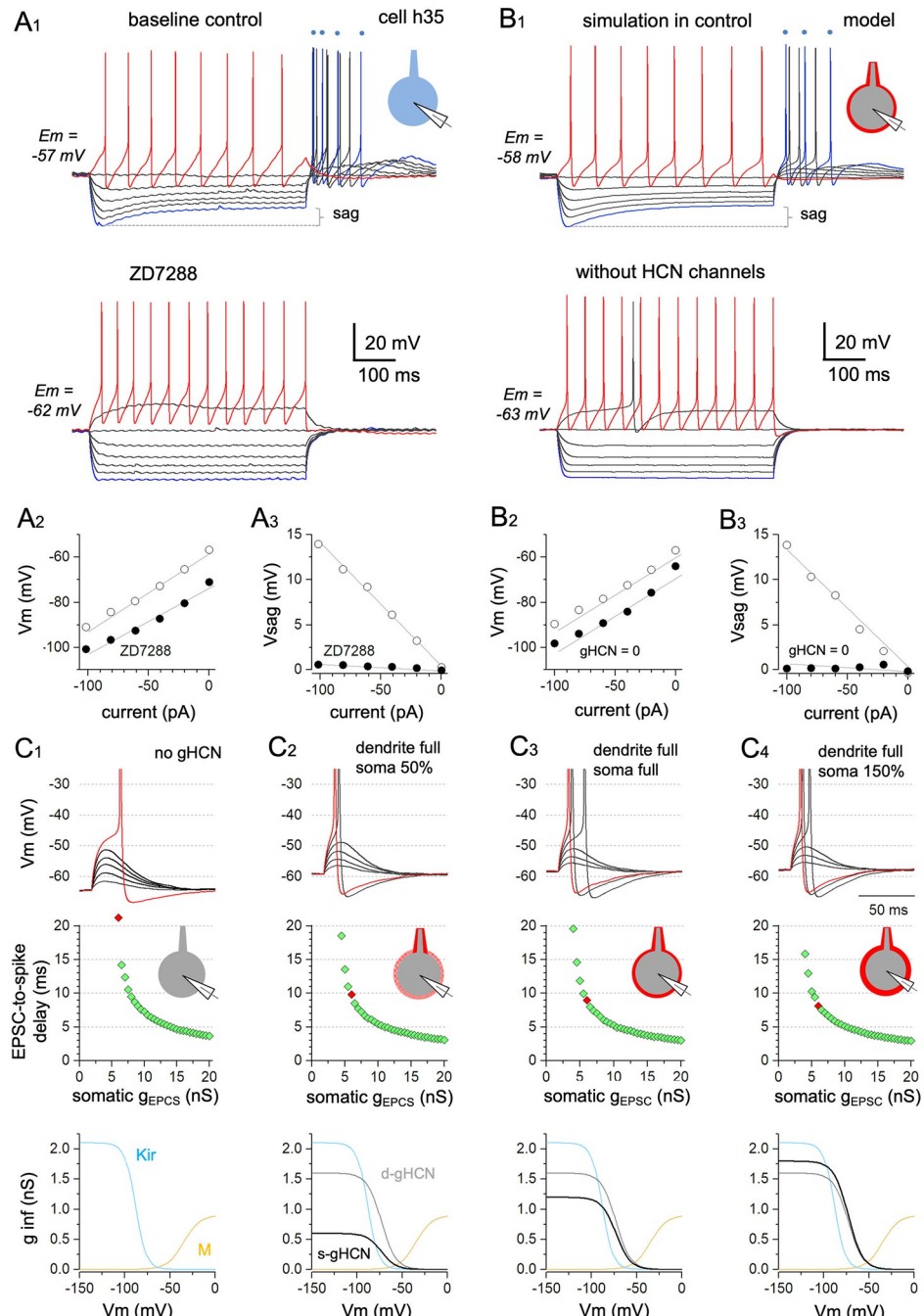

**Fig 6. A computational model demonstrates the effects of perisomatic HCN conductance modulation on human basket cell excitability and input–output function.** A three-compartmental model neuron with standard Hodgkin–Huxley-type spike-generating $Na^+$ and $K^+$ currents, slowly activating M-type $K^+$ current (M), inward-rectifying $K^+$ current (Kir), HCN current, and a physiological GABAergic autaptic conductance at the somatic compartment [35] reproduces the physiological behavior and EPSP–spike coupling of a human fast-spiking pv interneuron. (**A**) Membrane potential features recorded in human fast-spiking cell h35. (**A1**) Membrane potential responses to 500-ms current steps starting at −100 pA (blue trace) and reaching +20 pA (red traces) applied at Em (−57 mV). The bottom voltage traces are in the presence of the HCN channel blocker ZD7288 at Em = −62 mV. ZD7288 abolishes sag amplitude (sag) and rebound firing (blue dots show timing of rebound spikes for one membrane potential step). (Inset schematic indicates whole-cell recording from biological cell). (**A2**) Plot showing the peak negative voltage (Vm) against injected hyperpolarizing current-step amplitude at baseline (open symbols) and following ZD7288 wash-in (solid symbols). (**A3**) Voltage sag amplitude (Vsag) generated by incrementally increasing hyperpolarizing current steps (amplitude shown on abscissa) at baseline (open symbols). Black symbols illustrate blockade of sag by ZD7288.

(**B**) Membrane potential features of the h35 cell-based computational model. (**B1**) Top: The computational model closely replicates the electrophysiological response features of fast-spiking interneuron h35. Bottom: Removing HCN channels abolishes sag amplitude (sag) and rebound firing (indicated on top in one trace by blue dots) and reproduces the negative shift in Em. (Inset schematic indicates simulation with model cell.) (**B2**) Plot of peak negative voltage (Vm) versus injected current from the resting membrane potential for the computational model based on cell h35 with physiological gHCN activity (open symbols) at soma and in 200 μm-long primary dendrite, and when gHCN = 0 nS (black symbols). (**B3**) Sag amplitude (Vsag) during hyperpolarizing current steps with (open) and without gHCN (black symbols) at cell soma and in primary dendrite (see Methods and S2 Table for model cell details). (**C**) Somatic EPSP-evoked action potential time lag for the model with different HCN conductances (gHCN). (**C1**) EPSPs and EPSP-evoked action potentials in the model cell primary dendrite (200 μm long) and soma when gHCN = 0. Top: Superimposed traces show EPSPs elicited in the model cell soma by EPSCs of incrementally increasing conductance (gEPSC) from 1 to 6 nS. Red: An EPSP–spike waveform is evoked at 6 nS. Resting membrane potential is set at −62 mV. Middle: Plot showing EPSP-evoked spike delay measured from EPSC onset (EPSC-to-spike delay) as a function of somatic EPSC conductance (gEPSC) strength (abscissa) from 0.5 nS to 20 nS. Red dot indicates the EPSP–spike response evoked by a 6-nS gEPSC. Inset schematic indicates the generation of somatic EPSCs in the total somatodendritic compartment when gHCN = 0. Bottom: Curves show steady-state activation profile of the M-current (brown) and Kir-current (blue) conductances included in the models producing the voltage responses above. (**C2**) EPSP and EPSP–spike waveforms at physiological gHCN in the primary dendrite (200 μm) and at 50% of the somatic gHCN estimated for cell h35. Schematic inset shows full HCN at the dendrite and weak gHCN around the soma. Top: Traces show EPSP and EPSP–spike responses to EPSCs evoked by conductances from 1 nS to 6 nS at Em = −59 mV. Note action potentials are evoked at 5 nS and 6 nS (red). Middle: Spike delay changes with incremental increases in gEPSC. Red dot = 6 nS. Bottom: Activation profiles of the simulation showing M- and Kir- conductances and gHCN separately for dendritic (gray, d-gHCN) and somatic (black, s-gHCN) compartments. (**C3**) EPSPs and EPSP–spikes at full physiological gHCN estimated in the primary dendrite and soma (indicated in schematic inset with full HCN at dendrite and around soma). Top: EPSPs and EPSP–spike responses elicited at Em = −58 mV. Action potentials are evoked by the three input strengths (red = 6 nS). Middle: Spike delay changes with incremental increases in gEPSC. Red dot = 6 nS. Bottom: M- and Kir- conductances and full gHCN in both dendritic (gray) and somatic (black) compartment. (**C4**) EPSPs and EPSP–spike responses at full dendritic gHCN and 150% of h35 cell somatic gHCN (shown in schematic inset with strengthened perisomatic staining). Top: EPSPs and spikes elicited at Em = −57 mV. Action potentials are generated by the three input strengths (red = 6 nS). Middle: Spike delays at different gEPSC values (Red dot = 6 nS). Bottom: M-, Kir-, and HCN conductances with full dendritic gHCN (gray) and pronounced somatic gHCN (black) strength. The underlying data for Fig 6A–6C are available in S1 Data.

during hyperpolarization to −90 mV. We then simulated these responses (Fig 6B1–6B3, summarized in S2 Table) to obtain current–voltage relationships (passive and active) as well as sag amplitude, kinetics, and voltage dependence matching those of cell h35 under both baseline conditions and in the presence of ZD7288. Indeed, these simulations reproduced the membrane potential changes, firing responses to depolarizing current steps, sag amplitude voltage dependence, and rebound firing following hyperpolarizing steps observed in cell h35 (Fig 6A1 and 6B1).

Simulation showed that the activation of gHCN, which is responsible for the somatic voltage sag as described above, also leads to a depolarizing shift of Em by 5 mV (Fig 6A1, 6B1, and 6C1–6C4). We then investigated the contribution of gHCN to spike generation by EPSC input, generated by a synaptic glutamatergic-type conductance (0.5 nS to 20 nS), at the soma. Similar to our dynamic clamp experiments, this generated EPSPs to produce shortest spike delay close to 3 ms (Fig 6C). Large monosynaptic EPSCs with conductances between 5 and 15 nS have been reported in L2/3 human basket cells [11,13,19], and the EPSP–spike responses evoked in this conductance range were sensitive to perisomatic gHCN. Our simulation revealed that perisomatic gHCN reduces the EPSC amplitude required to elicit spiking at the resting membrane potential (Fig 6C) when gradually increased from 0 nS to partial somatic gHCN (50% of that measured in cell h35), physiological (100% of that measured from cell h35), and up to 150%. The effect of physiological HCN activity is shown by the difference in response between Fig 6C1 (total absence of gHCN in cell h35) and Fig 6C3 (simulation replicating physiological gHCN as shown in Fig 6A1 and 6B1). Finally, we found that the simulated EPSC-to-spike delay became progressively shorter as the somatic gHCN was increased from 0% to 50%, 100%, and 150% of the real somatic gHCN value at the soma of cell h35 (Fig 6C1–6C4).

Although HCN-mediated somatic current may attenuate the maximum EPSP amplitude by decreasing the input resistance (increasing the current leak) [43,46], the present model simulation indicates that this effect is more than compensated by Em depolarization and reduced τ (faster membrane kinetics). Consequently, an EPSP reaches higher peak depolarization [8,44] at a faster rate, thereby reducing the EPSC-to-spike delay.

## Discussion

Our results reveal the presence and demonstrate the physiological importance of an HCN channel-mediated membrane leak current in the soma of human fast-spiking GABAergic interneurons. These so-call fast-spiking basket cells are responsible for rapid synaptic inhibition in the neocortex, the brain area mediating the most complex cognitive and sensory processing tasks based on the precisely timed operation of neuronal networks [2,74,75]. We show that somatic HCN channel activity is a ubiquitous feature of these neurons in human neocortex but is absent or weak in fast-spiking basket cells of rodent neocortex [40–42,52–55] and further suggest that somatic gHCN is an essential adaptation to compensate for the slow (compared to rodent) passive membrane potential kinetics (kinetics measured in the absence of voltage sensitive ion channels such as HCN) of human basket cells stemming from low persistent ion leakage.

Unlike mouse basket cells, the human type exhibits a clear somatic sag potential, a hallmark of perisomatic HCN channels [21,43], and we further confirmed the presence of somatic HCN channels by applying the blocker ZD7288 in whole-cell patch-clamp experiments, by recording from soma-derived outside-out patches, and by high-resolution immunofluorescence staining. In addition, we also showed that both HCN1 and HCN2 channel isoforms are heavily localized at the cell soma membrane, similar to Kv3.1 potassium channels, which are characteristically localized at the pv cell perisomatic membrane [76–78]. Blockade experiments demonstrated that HCN channels in human cells depolarize the somatic membrane potential, accelerate membrane potential kinetics, and shorten the lag from EPSP induction to action potential generation [8,43,44]. Further, analysis of a computational model demonstrated that physiological level of somatic HCN channels can efficiently speed up the membrane potential kinetics of human pv-expressing basket cells [70].

### Perisomatic HCN channels in human fast-spiking cells ensure their "fast in–fast out" properties

Somatic HCN channels in human pv neurons can be regarded as an evolutionary adaptation that compensates for inherently slow passive membrane kinetics, which is inadequate for fast-spiking cells. Indeed, basket cells in the human neocortex exhibit sluggish membrane potential kinetics at the resting potential when HCN channels are blocked. This, in turn, increases the delay between EPSP input and spike output [22,35,36]. This slower EPSP–spike coupling compared with that of rodent basket cells results, at least partly, from a lower density of persistently open membrane ion channels at the resting potential and concomitantly higher resting input resistance, which leads to slower membrane potential kinetics [40,54,69,79–81]. Our experiments and simulations revealed that robust expression of voltage-sensitive HCN channels at the human basket cell soma membrane can increase membrane ion leak at the resting potential and thus accelerate EPSP–spike transformation. In fact, these channels ensure that human cell EPSP–spike transformation reaches nearly the same speed as that in mouse pv neurons. Thus, in human fast-spiking neurons, perisomatic HCN channels appear to act as a mechanism ensuring rapid and reliable EPSP–spike coupling [82]. The neuronal input–output rate is a critical factor for determining the processing speed of neuronal networks and, ultimately, the

computational power of the neocortex [75,83,84]. Although numerous mechanisms have evolved in mammalian neurons to enhance the neuronal input–output rate [8,17,20,38,85–87], there are special advantages to HCN channel expression, most notably the capacity to rapidly fine-tune somatic excitability and electrophysiological kinetics through regulation by neuro-modulators and intracellular second messengers.

## Human and mouse neurons use distinct ion channel combinations to set membrane excitability and kinetics at the soma

Our studies using the HCN blocker ZD7288 demonstrated clear differences in somatic input resistance and transmembrane ionic conductance between human and mouse fast-spiking neurons, in line with previous studies showing that human neurons in general have higher somatic input resistances than corresponding rodent neurons [22,35,69]. Why do rodent neurons at rest show a relatively large persistent conductance mediated by transmembrane potassium [52] and chloride leak [88], whereas in human neurons, the persistent ion leak is smaller and supplemented in the resting state by somatic HCN channels? One possible explanation is that this may allow for faster and more efficient regulation of somatic excitability and resting membrane potential in the human neurons. Our results and previous studies demonstrate that modulation of HCN channel activity or expression at the soma can powerfully regulate intrinsic excitability and firing fidelity [42,44,47,52,89–90]. In some neuron types, HCN channel activity can be rapidly up- or down-regulated by second messenger cascades initiated by neuronal excitation and activity [8,45,71]. Consequently, the intrinsic excitability state of a neuron can shift from highly responsive to input when gHCN is large to less responsive and slow to fire when HCN channel activity is low, and the membrane time constant is slow and the membrane is hyperpolarized. As demonstrated here by our modeling study, switching between such neuronal states can be effectively induced in human fast-spiking neurons through modulation of perisomatic HCN channel activity.

## Perisomatic HCN channels may have a general function in ensuring rapid membrane potential kinetics in human and other primate neurons

Our analysis of human tissue sources indicated that patient clinical condition, sex, or brain region of origin was not significantly associated with somatic HCN channels in human neurons. Their presence is ubiquitous across the neocortex and throughout life in the human brain. Moreover, the two key characteristics governing human fast-spiking basket cell excitability, high input resistance at the cell soma and a large HCN channel-mediated somatic voltage sag, have also been reported in fast-spiking interneurons of macaque monkey neocortex [40,41,54], suggesting that small persistent ion leak with robust HCN channel activity is a common adaptation of fast-spiking neurons in primates [64]. Human cortical pyramidal cells also exhibit a stronger sag potential and higher somatic HCN expression than rodent cortical pyramidal cells [21,47]. Thus, it appears that multiple human cortical neuron types employ robust somatic HCN channels.

In human neurons, somatic HCN channels shorten the spike lag in two ways. First, they directly reduce the somatic membrane time constant, and, second, they depolarize the membrane potential, which compensates for the greater leakage conferred by HCN channel opening and thereby maintains or increases spike probability as suggested by our model simulation at different gHCN levels. As a result, somatic HCN activity allows the cell to reach the action potential firing threshold faster and with higher probability. In addition, the deactivation of HCN channels during an EPSP speeds up the EPSP repolarization and this active property can

help to restrict spike initiation to the early phase of EPSPs [91]. Overall, somatic HCN channel-mediated leak current is well suited to facilitate EPSP–spike coupling in human neurons.

As real recordings and simulations show, without the somatic HCN conductance, EPSP-to-spike transformation would be at least 1 ms slower in the fast-spiking inhibitory neurons of human neocortex. In neuronal networks with multiple synapses separating excitatory and inhibitory neurons, even a small increase in single neuron transfer rate conferred by HCN channels could have a substantial cumulative effect on overall processing speed [17,84,92].

## Conclusions

Mammals have evolved numerous circuit, cellular, and molecular adaptations to enhance the speed and fidelity of cortical information processing, and it is likely that human-specific neuronal features contribute both to our greater cognitive capacity and susceptibility to various neurodegenerative and neuropsychiatric disorders. Identifying such species-specific differences may help explain uniquely human cognitive capability and provide novel therapeutic targets for disease treatment.

## Methods

### Ethics statement and licenses

All procedures were approved by the University of Szeged Ethics Committee and Regional Human Investigation Review Board (ref. 75/2014) and conducted in accordance with the tenets of the Declaration of Helsinki. Written informed consent for studies on excised human cortical tissue was obtained from all patients prior to surgery.

Clip art images (rodent and human silhouette insets) were downloaded from online open clip art library http://clipart-library.com and images were further modified in color and relative dimensions for final figures.

### Human brain slices

Neocortical slices were prepared from samples of frontal, temporal, or other cortex removed for the surgical treatment of deep-brain targets. Patients ranged from 20 to 82 years of age, and samples were acquired from both the left and right hemispheres of males and females. Anesthesia was induced with intravenous midazolam (0.03 mg/kg) and fentanyl (1 to 2 μg/kg) following bolus intravenous injection of propofol (1 to 2 mg/kg). Patients also received 0.5 mg/kg rocuronium to facilitate endotracheal intubation. Patients were ventilated with a 1:2 $O_2/N_2O$ mixture during surgery, and anesthesia was maintained with sevoflurane. Following surgery, the resected tissue blocks were immediately immersed in an ice-cold solution containing (in mM) 130 NaCl, 3.5 KCl, 1 $NaH_2PO_4$, 24 $NaHCO_3$, 1 $CaCl_2$, 3 $MgSO_4$, and 10 D(+)-glucose aerated with 95% $O_2$/5% $CO_2$ within a glass container. The container was then placed on ice inside a thermally isolated box and immediately transported from the operating room to the electrophysiology laboratory with continuous 95% $O_2$/5% $CO_2$ aeration. Slices of 350 μm thickness were prepared from the tissue block using a vibrating blade microtome (Microm HM 650 V) and then incubated at 22°C to 24°C for 1 h in slicing solution. The slicing solution was gradually replaced with a recording solution (180 mL) using a pump (6 mL/min). The contents of the recording solution were identical to the slicing solution except that it contained 3 mM $CaCl_2$ and 1.5 mM $MgSO_4$.

### Drug

The HCN channel blocker ZD7288 (Sigma-Aldrich, Budapest, Hungary) was diluted in physiological extracellular solution and applied by wash-in.

### Fast-spiking basket cells in mouse brain slices

Transversal slices (350 μm) were prepared from the somatosensory cortex and in frontal cortex of 4- to 6-week-old heterozygous B6.129P2-Pvalbtm1(cre)Arbr/J mice (Jackson Laboratory, stock 017320, B6 PVcre line) crossed with Ai9 reporter line to express the tdTomato fluorophore in pv GABAergic neurons to assist in cell selection. Cell identity was tentatively confirmed electrophysiologically by fast spike kinetics and high-frequency nonaccommodating firing in response to 500-ms suprathreshold depolarizing pulses and subsequently confirmed as basket cells by staining with streptavidin conjugated to Alexa 488 (1:2,000, Jackson ImmunoResearch, West Grove, PA, USA) and examination under epifluorescence microscopy.

### Electrophysiology

Recordings were performed in a submerged chamber perfused at 8 mL/min with recording solution maintained at 36˚C to 37˚C. Cells were patched under visual guidance using infrared differential interference contrast video microscopy and a water-immersion 20× objective with additional zoom (up to 4×). All cells experiments were performed within 30 min from entering to whole cell mode. Micropipettes (5 to 8 MΩ) for whole-cell patch-clamp recording were filled with intracellular solution containing (in mM) 126 K-gluconate, 8 NaCl, 4 ATP-Mg, 0.3 Na$_2$–GTP, 10 HEPES, and 10 phosphocreatine (pH 7.0 to 7.2; 300 mOsm) and supplemented with 0.3% (w/v) biocytin for subsequent staining with Alexa 488–conjugated streptavidin. Recordings were performed with a Multiclamp 700B amplifier (Axon Instruments) and low-pass filtered at a 6 to 8 kHz cutoff frequency (Bessel filter). Series resistance and pipette capacitance were compensated in current-clamp mode, and pipette capacitance was compensated in voltage-clamp mode. In voltage-clamp recordings, the effect of access resistance (measured first in current-clamp) on the nominal clamping potential reading was calculated and corrected during data analysis. The resting membrane potential (Em) was recorded starting 1 to 3 min after patch rupture. Membrane potential values were not corrected for liquid junction potential error. The somatic leakage conductance (Gleak) was measured in voltage-clamp mode as the ratio of evoked current to voltage step amplitude (−10 mV for 10 ms) according to Ohm's law as follows: Gleak (nS) = clamping current amplitude at the end of the voltage step (pA)/voltage-clamp step amplitude (mV). Somatic input resistance was measured in current-clamp mode by measuring the current required to hyperpolarize the membrane to −90 mV peak potential (250 to 500 ms) from −70 mV. All parameters were measured from at least five traces. Outside-out somatic membrane recordings were conducted after whole-cell recordings by retracting the pipette under visual guidance using a microscope-associated camera. Bridge balance was set during experiments. Sag amplitudes were calculated from at least 5 steps in each configuration.

### Dynamic clamp

A software-based dynamic clamp system was employed to produce voltage waveforms simulating EPSPs. Current injections were calculated and delivered by Signal software (Cambridge Electronic Design, Cambridge, UK) through a Power1401-3A data acquisition interface (Cambridge Electronic Design) based on voltage signals of the electrode. We ran the dynamic clamp on a computer distinct from our experimental data acquisition system (recording cell membrane potential) to record and verify dynamic clamp conductance and output EPSCs. The

EPSP waveforms were produced by EPSCs with a decay time constant of 3 ms, reversal potential of 0 mV, and dynamic clamp input conductance range from 1.5 to 10 nS.

## Single-cell model

We constructed a model neuron incorporating electrophysiological parameters derived from experimental findings to assess the effects of somatic HCN channels on the input–out responses of basket cells. The three-compartmental model neuron including cylindrical dendritic and axonal compartments was based on the formalism described previously [70]. The model consisted of a somatic compartment in addition to one axonal and one dendritic cable compartment, both 200 μm in length and 1.2 μm in diameter. Cable parameters were axon longitudinal resistance = 1.0 MΩ per μm and capacitance = 12 fF/μm$^2$ for both the axon and dendrite (also referred to as the "primary dendrite"). The somatic compartment had an ohmic resistance of 400 MΩ, capacitance of 12 pF, and the leakage current reversal potential of −62 mV. In addition, the model incorporated 5 voltage-dependent currents: transient Na$^+$ current, delayed rectifying K$^+$ current (Kd), HCN current, slowly activating M-type K$^+$ current, and inward-rectifying K$^+$ current (Kir). Parameters such as rheobase and voltage sag were chosen to replicate basic response properties and adjusted using experimental data to attain the best match with recorded voltage traces. All intrinsic voltage-dependent currents were calculated according to the Eq 1:

$$I_i = g_i m_i^p h_i (E_i - V),$$

where $i$ represents the individual current type, $g_i$ is the maximal conductance of the current, $m_i$ is the activation variable, $p$ is the exponent of the activation term, $h_i$ is the inactivation variable (either first-order or absent), and $E_i$ is the reversal potential. Activation ($m$) and inactivation ($h$) kinetics were modeled according to the Eq 2:

$$\frac{dx}{dt} = \frac{x_\infty(V) - x}{\tau_x(V)},$$

where $x$ represents $m$ or $h$, and voltage-dependent steady-state activation and inactivation are described by the sigmoid function Eq 3:

$$x_\infty(V) = \frac{1}{2} + \frac{1}{2} \tanh\left(\frac{V - V_{x,1/2}}{V_{x,sl}}\right).$$

The midpoint $V_{x,1/2}$ and slope $V_{x,sl}$ parameters of the sigmoids and the other kinetic parameters are shown in S2 Table. Time constants ($\tau$) of activation and inactivation were modeled as bell-shaped functions of the membrane potential ($V$) according to the Eq 4:

$$\tau_x(V) = \left(\tau_{x,max} - \tau_{x,min}\right)\left[1 - \tanh\left(\frac{V - V_{\tau x,1/2}}{V_{\tau x,sl}}\right)^2\right] + \tau_{x,min}.$$

The voltage-dependent Na$^+$, and K$^+$ Kd- and M-currents were assigned to the axonal and somatic compartments, while the HCN and Kir currents were included in the somatic and dendritic compartments. The model was tested using a current-step protocol matching the one used for biological neurons (0.5 s duration current steps starting at −100 pA and increasing in increments of +20 pA). Membrane resistance, time constant, voltage sag, and other parameters were calculated from the voltage responses and compared to those observed in representative human basket cell. In addition to the intrinsic voltage-dependent currents, we

included a chemical GABAergic autaptic connection from the axon to the somatic compartment of the model as autaptic inhibition is a regular feature of human L2/3 neocortical pv interneurons and has a clear impact on the hyperpolarizing phase of action potential waveform [35]. The parameters of the autaptic connection were as follows: conductance = 2.5 nS, conductance onset delay to action potential peak = 1 ms; time to peak = 0.8 ms; current decay time constant = 10 ms; activation threshold = −32 mV; slope of the activation function = 20 mV; reversal potential of the autaptic current = −72 mV.

Autaptic current was described using a first-order kinetics of transmitter release as defined in Eq 5:

$$I_{syn} = g_{syn}S(E_{syn} - V),$$

where $S$ is the instantaneous autaptic activation term yielding the following differential Eq 6:

$$\frac{dS}{dt} = \frac{S_\infty(V_{pre}) - S}{\tau_{syn}(1 - S_\infty(V_{pre}))}.$$

The steady-date autaptic activation term depends on the cell membrane potential as defined in Eq7:

$$S_\infty\left(V_{pre}\right) = \tanh\left(\frac{V_{pre} - V_{th}}{V_{slope}}\right)$$

when $V_{pre} > V_{th}$. Otherwise $S_\infty(V_{pre}) = 0$. The term $V_{pre}$ denotes the membrane potential of the axon cable.

## Data analysis

Data were acquired using Clampex software (Axon Instruments), digitized at 35 to 50 kHz, and analyzed offline using pClamp (version 10.5, Axon Instruments), Spike2 (version 8.1, Cambridge Electronic Design), OriginPro (version 9.5, OriginLab Corporation), and Sigma-Plot (version14, Systat Software) softwares.

## Statistics

Results are presented as median with lower to upper quartile range (IQR) unless stated otherwise. Statistical significance was evaluated by Mann–Whitney U test, Wilcoxon signed-rank test, or ANOVA on Ranks (Kruskal–Wallis H test) as indicated using Sigma Plot. Correlations were tested using Pearson's or Spearman Rank Order method. A $P < 0.05$ was considered significant for tests. Parametric distribution was confirmed with the Shapiro–Wilk test as $P > 0.05$.

## Tissue fixation and cell visualization

Cells filled with biocytin from whole-cell patch-clamp recording were visualized using either Alexa 488–conjugated streptavidin (1:1,000) or Cy3-conjugated streptavidin (1:1,000) (both from Jackson ImmunoResearch). After recording, slices were immediately fixed in a solution containing 4% paraformaldehyde and 15% picric acid in 0.1 M phosphate buffer (PB; pH = 7.4) at 4˚C for at least 12 h and then stored at 4˚C in 0.1 M PB containing 0.05% sodium azide as a preservative. All slices were embedded in 20% gelatin and further cut into 60-µm thick sections in ice-cold PB using a vibratome (VT1000S, Leica Microsystems, Wetzlar, Germany). After sectioning, they were rinsed in 0.1 M PB (3 × 10 min), cryoprotected in 10% to

20% sucrose solution in 0.1 M PB, flash-frozen in liquid nitrogen, and thawed in 0.1 M PB. Slice sections were then incubated in 0.1 M Tris-buffered saline (TBS; pH 7.4) containing fluorophore-conjugated streptavidin for 2.5 h at 22°C to 24°C. After washing with 0.1 M PB (3 × 10 min), the sections were covered in Vectashield mounting medium (Vector Laboratories, Burlingame, CA, USA), placed under coverslips, and examined under an epifluorescence microscope at 20 to 60× magnification (Leica DM 5000 B).

Sections (60-μm thick) used for single-cell reconstruction were further incubated in a solution of conjugated avidin-biotin horseradish peroxidase (ABC; 1:300; Vector Labs) in TBS (pH = 7.4) at 4°C overnight. The enzyme reaction was visualized by the glucose oxidase-DAB-nickel method using 3′3-diaminobenzidine tetrahydrochloride (0.05%) as the chromogen and 0.01% $H_2O_2$ as the oxidant. Sections were further treated with 1% $OsO_4$ in 0.1 M PB. After several washes in distilled water, sections were stained with 1% uranyl acetate, dehydrated in an ascending series of ethanol concentrations, infiltrated with epoxy resin (Durcupan) overnight, and embedded on glass slides. Light microscopic reconstructions were conducted using the Neurolucida system with a 100× objective (Olympus BX51, Olympus UPlanFI). Images were collapsed in the z-axis for illustration.

### Immunohistochemistry

Free-floating sections were washed 3 times in TBS plus 0.3% Triton X (TBST) for 15 min at 22°C to 24°C and then transferred to a blocking solution of 20% horse serum in TBST for pv staining. All sections were incubated in primary antibodies diluted in TBST over three nights at 4°C and then in the appropriate fluorochrome-conjugated secondary antibody solution (1% blocking serum in TBST) overnight at 4°C. Sections were first washed in TBST (3 × 15 min), then washed in 0.1 M PB (2 × 10 min), and mounted on glass slides with Vectashield mounting medium (Vector Laboratories). The following primary antibodies were used for immunostaining of brain sections: mouse anti-pv (1:500, Swant, Switzerland, www.swant.com, clone: 235); rabbit anti-HCN1 (1: 500, MyBioSource), and rabbit anti-KV3.1 (1:500, Synaptic Systems, www.sysy.com). Immunolabeling was visualized for confocal microscopy using the following secondary antibodies: DAM Alexa 488–conjugated donkey antimouse (1:400, Jackson ImmunoResearch, www.jacksonimmuno.com) or DAM Cy3-conjugate donkey antimouse (1:400, Jackson ImmunoResearch). For dSTORM, secondary antibodies were DARb Alexa 647 donkey anti-rabbit (1:500, Abcam, www.abcam.com) and DaRb CF568 (1:500, Biotium). The immunoreactions were evaluated first using an epifluorescence microscope (Leica DM 5000 B) and then a laser scanning confocal microscope (Nikon Eclipse Ti-E).

### Confocal microscopy

Confocal images were captured using a Nikon C2+ confocal scan head attached to a Nikon Eclipse Ti-E microscope equipped with a high NA objective (Nikon CFI Apo TIRF 100XC Oil, NA = 1.49). The setup and data acquisition process were controlled by Nikon NIS-Elements 5.02 software, and images were postprocessed in MATLAB. The Nikon Laser Unit was used to set the wavelength and power of the following applied lasers: Sapphire 488 LP-200 (Pmax = 200 mW, Coherent, Santa Clara, CA, USA) for 488 nm excitation; Cobolt Jive (Pmax = 300 mW, Cobolt, Kassel, Germany) for 561 nm excitation; 2RU-VFL-P-300-647-B1 (Pmax = 300 mW, MPB Communications, Montreal, Canada) for 647 nm excitation.

### Line analysis of immunofluorescence intensity

Immunofluorescence images of pv-expressing neurons as well as fluorophore-labeled HCN1, HCN2, and Kv3.1 channels were captured using a Leica Stellaris-8 confocal microscope

system, with special care to avoid pixel saturation. Images were then analyzed offline using LAS X Life Science Microscope Software. Briefly, an analysis template with six radial fluorescence measuring lines was positioned on an individual pv-immunopositive cell soma so that the template center was approximately at the soma center and the end of each measuring line (length 8 μm, 15 pixels per μm) extended to the extracellular space. Thus, each line read pixel-level fluorescence intensity from the soma center to the edge (border of the extracellular space in two dimensions). The lines were positioned symmetrically at 60˚ angles. Fluorescent signals from pv immunoreactivity in human cells and genetically encoded tdTomato in mouse cells was used to define the intracellular region. Intracellular signals from line analysis are likely to originate from both cytoplasm and nucleus since we were not able to clearly discriminate these subcellular regions. For analysis, the membrane zone was defined exactly as a 1-μm-wide region with midpoint where pv (or tdTomato) fluorescence intensity reached half maximum (measured between 20% and 80% of signal minimum and maximum amplitude), while the extracellular region was defined from immediately outside this membrane zone. For measurement of fluorescence intensity values in each cell, readings from the six lines were first aligned and averaged as demonstrated in Fig 2A–2D, 2F and 2G. Average fluorescence intensity values for labeled HCN1 and HCN2 channels used to calculate ratios in Fig 2 were collected from 3-μm, 1-μm, and 1-μm line lengths across intracellular, membrane zone, and extracellular regions, respectively.

## dSTORM microscopy

Super-resolution dSTORM measurements were performed on a custom-made inverted microscope based on a Nikon Eclipse Ti-E frame with an oil immersion objective (Nikon CFI Apo TIRF 100XC Oil, NA = 1.49). Epifluorescence illumination was applied at an excitation wavelength of 647 nm. The laser intensity was set to 2 to 4 kW/cm$^2$ on the sample plane and controlled via an acousto-optic tunable filter. An additional laser (405 nm, Pmax = 60 mW; Nichia) was used for reactivation. A filter set from Semrock (including a Di03-R405/488/561/635-t1-25 × 36 BrightLine quad-edge super-resolution/TIRF dichroic beam-splitter and FF01-446/523/600/677-25 BrightLine quad-band bandpass filter, and an additional AHF 690/70 H emission filter) was inserted into the microscope to separate excitation from emission wavelengths. Images of individual fluorescent dye molecules were captured by an Andor iXon3 897 BV EMCCD camera (512 × 512 pixels with 16 μm pixel size) with the following acquisition parameters: exposure time = 20 ms; EM gain = 200; temperature = −75˚C. Typically, between 20,000 and 25,000 frames were captured from a single region of interest. The Nikon Perfect Focus System was used to keep the sample in focus during measurements. High-resolution images were reconstructed with the rainSTORM localization software (http://laser.cheng.cam. ac.uk/wiki/index.php/Resources). Spatial drift introduced by either mechanical movement or thermal effects was analyzed and reduced using an autocorrelation-based blind drift correction algorithm. Background autofluorescence was calculated as the weighted moving average of neighboring pixel values and subtracted from each frame before the localization step. Using this subtraction method, pixel values within the noise level carried more weight because they do not belong to a fluorescent burst of a switching fluorophore. The detected localizations were filtered based on the fitting parameters (e.g., sigma values, residue, and intensity). The pixel size of the final pixelized super-resolution images (typically 20 nm) was set based on the localization precision and localization density. The dSTORM experiments were conducted in a GLOX switching buffer, and the sample was mounted onto a microscope slide. The imaging buffer was an aqueous solution diluted in PBS containing the GluOx enzymatic oxygen scavenging system, 2,000 U/mL glucose oxidase (Sigma-Aldrich, catalog number: G2133-50KU),

40,000 U/mL catalase (Sigma-Aldrich, catalog number: C100), 25 mM potassium chloride (Sigma-Aldrich, catalog number: 204439), 22 mM tris(hydroxymethyl)aminomethane (Sigma-Aldrich, catalog number: T5941), 4 mM tris(2-carboxyethyl)phosphine (TCEP) (Sigma-Aldrich, catalog number: C4706) with 4% (w/v) glucose (Sigma-Aldrich, catalog number: 49139), and 100 mM β-mercaptoethylamine (MEA) (Sigma-Aldrich, catalog number: M6500). The final pH was set to 7.4.

## Supporting information

**S1 Data. Numerical dataset for each figure of this manuscript.**
(XLSX)

**S1 Table. Details of human fast-spiking layer 2/3 basket cells analyzed in this study.** Part I includes 72 fast-spiking cells studied for systematic sag amplitude in the whole-cell mode (evoked by hyperpolarizing steps to −90 mV from −70 mV). From left to right: experiment code, patient sex (M = male, F = female), patient age in years (yrs), resected neocortical tissue hemisphere (left or right), cortical area resected, patient primary diagnosis for surgery, sag amplitude (Ih sag, mV) measured during voltage steps from −70 mV to −90 mV, width of action potential escape current inward component (in ms) measured at the onset point of the fast depolarizing current [35], resting membrane potential (Em), firing accommodation index measured as the ratio of spike numbers within the 400–500 ms and 0–100 ms time windows during a 500-ms depolarizing pulse inducing high-frequency but not maximal firing [35,62], "accommodation Hz" indicates action potential firing frequency during the 0–100 ms time window of the depolarizing pulse used for accommodation index calculation; immunohistochemistry shows details of pv immunoreactivity (pv+ = pv-immunopositive in soma-dendrite or axon; nonconclusive means unsuccessful pv immunostaining, no recovery means unsuccessful cell visualization with biocytin. Part II summarizes the patient data for the resected tissues used in immunohistochemical studies of HCN1, HCN2, and Kv3.1 (Figs 2 and 3). Table shows tissue block identification code, patient data, and the immunochemical (IHC) study performed. S1 Data shows line analysis intensity values for all cells studied for HCN1 expression (8 patients) and HCN2 expression (6 patients). Part III summarizes the data from 13 fast-spiking human cells examined by both whole-cell and outside the out patch recordings and the corresponding patient data. Indicated as o-o patch cells 1 to 13 and studied with hyperpolarizing step to −90 mV from −60 mV.
(DOCX)

**S2 Table. Voltage-dependent current parameters for the generic human basket cell model.** Columns from left: Current = Transmembrane current type and conductance (g, nS) in soma, dendrite, and axon. E (mV) = reversal potential of current; $p$ = exponent of activation term.
(DOCX)

## Acknowledgments

We thank Leona Mezei and Drs Gábor Molnár and Katalin Kocsis for technical assistance.

## Author Contributions

**Conceptualization:** Karri Lamsa.

**Data curation:** Viktor Szegedi, Emőke Bakos, Szabina Furdan, Bálint H. Kovács, Miklós Erdélyi.

**Formal analysis:** Viktor Szegedi, Szabina Furdan, Dániel Varga, Attila Szücs.

**Funding acquisition:** Karri Lamsa.

**Investigation:** Viktor Szegedi, Emőke Bakos, Szabina Furdan, Bálint H. Kovács, Dániel Varga, Attila Szücs.

**Methodology:** Bálint H. Kovács, Miklós Erdélyi.

**Project administration:** Karri Lamsa.

**Resources:** Miklós Erdélyi, Pál Barzó, Gábor Tamás.

**Software:** Bálint H. Kovács, Attila Szücs.

**Supervision:** Miklós Erdélyi, Karri Lamsa.

**Validation:** Viktor Szegedi.

**Visualization:** Dániel Varga, Attila Szücs, Karri Lamsa.

**Writing – original draft:** Karri Lamsa.

**Writing – review & editing:** Karri Lamsa.

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
