## [Editor Report · Decision Letter 0]

25 Mar 2022

***HAVE YOU ASSIGNED THE AE? IF NOT CANCEL THIS DECISION AND GO BACK!***

***REMEMBER TO CLOSE ALL REDUNDANT AE DISCUSSIONS***

Dear Dr Lämsä, 

Thank you for submitting your manuscript entitled "Somatic HCN channels accelerate the input–output kinetics of human cortical

fast-spiking interneurons" for consideration as a Research Article by PLOS Biology.

Your manuscript has now been evaluated by the PLOS Biology editorial staff [as well as by an academic editor with relevant expertise] and I am writing to let you know that we would like to send your submission out for external peer review.

Once your full submission is complete, your paper will undergo a series of checks in preparation for peer review. Once your manuscript has passed the checks it will be sent out for review. To provide the metadata for your submission, please Login to Editorial Manager (https://www.editorialmanager.com/pbiology) within two working days, i.e. by Mar 27 2022 11:59PM.

If your manuscript has been previously reviewed at another journal, PLOS Biology is willing to work with those reviews in order to avoid re-starting the process. Submission of the previous reviews is entirely optional and our ability to use them effectively will depend on the willingness of the previous journal to confirm the content of the reports and share the reviewer identities. Please note that we reserve the right to invite additional reviewers if we consider that additional/independent reviewers are needed, although we aim to avoid this as far as possible. In our experience, working with previous reviews does save time. 

If you would like to send previous reviewer reports to us, please email me at kdickson@plos.org;dickson.kris@gmail.com to let me know, including the name of the previous journal and the manuscript ID the study was given, as well as attaching a point-by-point response to reviewers that details how you have or plan to address the reviewers' concerns. 

Given the disruptions resulting from the ongoing COVID-19 pandemic, please expect some delays in the editorial process. We apologise in advance for any inconvenience caused and will do our best to minimize impact as far as possible.

Kind regards,

Kris

Kris Dickson

PLOS Biology

kdickson@plos.org;dickson.kris@gmail.com

---

## [Decision Letter · Decision Letter 1]

30 Apr 2022

Dear Dr Lämsä,

Thank you very much for submitting a revised version of your manuscript "Somatic HCN channels accelerate the input–output kinetics of human cortical fast-spiking interneurons" for consideration as a Research Article at PLOS Biology. This revised version of your manuscript has been evaluated by the PLOS Biology editors, the Academic Editor and reviewers. Reviewer 1 also saw the prior version of your manuscript (PBIOLOGY-D-21-01590). As the other two original reviewers were not available, two new reviewers with relevant expertise were brought in to evaluate this new version of your study. 

As you will see, Reviewer 1 appreciated the additional experiments that were added in this new version of your manuscript. However, they still felt that a number of issues are still present that preclude strong conclusions being drawn. Many of these concerns were also raised by the two new reviewers as well, including the need for additional data to support the core claim that differences in somatic HCN channel density underlie the effects you see. In light of the reviews we've received (below), we will not be able to accept the current version of the manuscript. However, given the general enthusiasm still being expressed for the potential conclusions of this study and the fact that the reviewers give clear advice on what they feel is still needed to bolster your findings, we would welcome re-submission of a much-revised version that takes into account all of the reviewers' comments. We would be unlikely to allow a third round of revision, so we strongly encourage you to do your best to fully address all of the concerns that have been raised in this round of review. 

Please note that we cannot make any decision about publication until we have seen the revised manuscript and your response to the reviewers' comments. Your revised manuscript is also likely to be sent for further evaluation by these reviewers.

We expect to receive your revised manuscript within 3 months. However, if you feel that you would require additional time to fully address the remaining concerns, please email the PLOS Biology office (plosbiology@plos.org) if you would like to request an extension. You can also reach out to me directly if you have any questions (kdickson@plos.org). At this stage, your manuscript remains formally under active consideration at our journal; please notify us by email if you do not intend to submit a revision so that we may end consideration of the manuscript at PLOS Biology.

**IMPORTANT - SUBMITTING YOUR REVISION**

*Re-submission Checklist*

*Published Peer Review*

*PLOS Data Policy*

*Blot and Gel Data Policy*

Sincerely,

Kris

Kris Dickson

Neurosciences Senior Editor/Section Manager

PLOS Biology

kdickson@plos.org

REVIEWS:

Reviewer's Responses to Questions

PLOS authors have the option to publish the peer review history of their article (what does this mean?). If published, this will include your full peer review and any attached files.

Reviewer #1: No

Reviewer #2: No

Reviewer #3: No

Reviewer #1: In comparison to the previous version of the manuscript, the current version has been significantly improved. Specifically, the authors have conducted additional experiments to support that HCN channels are expressed in human parvalbumin-expression basket cell (PV+-BC) somata. Furthermore, they have performed analyses to demonstrate that the targeting of HCN channels to human PV+-BC somata is independent of a range of different factors. Collectively, these results suggest that somatic HCN channels may be a widespread feature of human cortical PV+-BCs. However, the current manuscript still has several issues.

Major 

(1) In nucleated patches from human PV+-BCs, a hyperpolarizing current pulse evokes a sag in the voltage response (Figure 1D). The authors use this result as the electrophysiological evidence to support the idea of somatic expression of HCN channels in human PV+-BCs, but this experiment has a weakness. Although activation of HCN channels can create a sag at the beginning of long-lasting hyperpolarizations, HCN channels are not the only mechanism that can give rise to the sag response. In essence, the sag can also be produced by the closing of slow and non-inactivating K+ channels, for example, KCNQ channels. Consequently, additional pharmacological experiments with ZD7288 are required to confirm that HCN channels can account for the sag response in nucleated patches from human PV+-BCs.

(2) Examples in Figure 2 qualitatively suggest that HCN channel expression pattern in human PV+-BCs is different from that of mouse interneurons. However, a quantitative analysis to compare the density of HCN1 immunosignal in human PV+-BC somata with its mouse counterpart is lacking.

(3) The authors have described that HCN channels can accelerate the conversion of EPSPs to action potentials (APs) by reducing the membrane time constant (end of the first paragraph on page 10) and by depolarizing the resting membrane potential (page 14). However, there is one additional factor that has not been mentioned. The deactivation of HCN channels during an EPSP speeds up the EPSP repolarization (Magee JC, 1998, J Neuroscience), and this active property can help to restrict AP initiation to the early phase of EPSPs. This mechanism should be included in the discussion. 

(4) My previous comment about the interplay between h- and M-currents during the EPSP-to-AP conversion of human PV+-BCs has not been sufficiently addressed. The minimum, which the authors should do, is to cite the relevant article and discuss this. Furthermore, I do not think that supplementary figure 2 can help us to address this issue because EPSP-to-AP conversion has not been analyzed in this figure. 

Minor 

(1) My previous comment 'The authors used the dynamic-clamp method to simulate excitatory synaptic inputs to PV+-BCs. A dynamic-clamp system converts a defined conductance to currents based on the value of the membrane potential. Consequently, the output of a dynamic-clamp system is a current rather than a conductance' has not been sufficiently addressed (last paragraph on page 9, Figure 4 legend on page 19 of the current manuscript). In a dynamic-clamp experiment, a signal applied to the input of the dynamic-clamp system is treated as a conductance instead of an EPSC. Furthermore, siemens cannot be used as the unit to describe the magnitude of EPSCs (Figure 4B2 legend). 

(2) There is a mismatch between color codes in Figures 1A1 and 1B2. In Figure 1A1. The biocytin signal is represented by red color, and the parvalbumin signal is represented by green color. In Figure 1B2, the color code is the opposite. 

(3) When describing conductance and capacitance values of the model (page 25), please normalize the values with the membrane area of the model. This facilitates the comparison of the parameters in this paper with the result from a previous study, which shows that human cortical pyramidal neurons display unique passive membrane properties in comparison to their rodent counterparts (Eyal G. et al., 2016, eLife). Similarly, the magnitude of active conductances in the model should be described by using conductance density (Supplementary Table 2). 

(4) Please describe which currents 'D' and 'A' in Supplementary table 2 represent. Are these D-type and A-type K+ currents? Why did the authors choose different reversal potential values for different types of K+ current (Supplementary table 2)?

Reviewer #2: Summary:

The authors, through electrophysiological recordings as well as immunohistochemical staining of parvalbumin positive interneurons (PVIs), show that human PVIs express more HCN channels in their somatic cellular membrane than the same cell type in the mouse cortex. The authors further show that in human PVIs HCN channels decrease the membrane time constant and thus facilitate faster action potential output in response to EPSP input close to AP threshold. These experimental results are corroborated by a computational single cell model with parameters based on one of the recorded cells, in which modification of HCN conductance produces similar effects as observed in the electrophysiological measurements.

In general, the study provides convincing evidence for a difference in HCN channel expression between human and mouse cortical PVIs. The resulting effects on membrane time constant and input-output delay are not unexpected but have interesting implications for the precision of inhibition in cortical circuits. However, some technical issues concerning the relation between sag measurements and underlying Ih conductances remain to be addressed.

Major Comments:

1. In figure 1, the authors show a wide range of sag amplitudes in the human PV interneurons. It would be interesting to discuss if this range is correlated with other parameters (cell size, baseline input resistance, dendritic/axonal morphology, etc.) since there are multiple sub-classifications of PV interneurons in the cortex.

2. One of the key difficulties in comparing sag values between different cell types is to determine whether any differences are due to differences in intrinsic Ih properties, such as peak conductance, steady-state voltage-dependent gating, or Ih kinetics, versus differences in extrinsic factors, such as membrane potential, input resistance of the cell, or the cell's membrane time constant. All of these can act as confounding factors. Although the authors attempt to control for some of these factors, there are some concerns. For example in Fig 1E, p5, 2nd paragraph, the authors compare sag in humans with a hyperpolarizing step to -91 mV with sag in mouse using a voltage step to -85 mV. This 6 mV difference can likely lead to a several-fold larger Ih conductance increase in human vs mouse, depending on the mid-point voltage of Ih activation. The nicest was the authors have of controlling for such effects is through the outside-out patch data they show in Fig 1D. Unfortunately this is only shown for human cells and even there the number of experiments and mean date are not provided. As another means of assessing Ih in a manner that will be less prone to influence of extrinsic factors, the authors could measure Ih under voltage clamp from the data of Fig 3, subtracting the current in the presence of ZD from the current in the absence of ZD. 

3. The logic of the experiment of Fig 4b,c examining the role of Ih in action potential firing is rather unclear. I would have thought the primary goal would be to ask how spike firing is altered without the presence of Ih, without altering other parameters that are indirectly affected by Ih. In contrast, the authors artificially manipulate the resting potential to keep it constant and decrease the size of the EPSC to compensate for the change in input resistance due to loss of Ih. Although this might be of interest to examine why loss of Ih may alter spike firing, it does not provide an accurate view of how loss of Ih may affect cell firing. The authors should therefore repeat the experiments with ZD7288 but keeping the same amplitude EPSC as in control and allowing the resting potential to hyperpolarize to its appropriate level. 

4. As ZD7288 may exert off target effects on sodium channels, the authors should also determine effects of threshold voltage and rheobase, both from a fixed holding potential (eg -70 mV) and from the actual resting potential measured with or without drug. 

5. Discussion paragraph 1: "In fact, these channels are necessary to reduce the EPSP-spike delay to roughly that of mouse basket cells." - This is a strong statement, given that figure 4C1 shows that even after ZD7288 the delay in mouse cells seems much shorter. The authors should tone down this conclusion.

6. Discussion paragraph 2: "Further, the absence of other leak conductances may reduce the energy expenditure required to maintain transmembrane ionic gradients." - I am not quite clear on how another leak conductance could facilitate processing speed as alluded to in this paragraph. In fact, the effects attributed to HCN channels in this study might best be approximated by less leak current in the soma, which would result in an overall higher input resistance and faster time constant as well. Therefore, the claim that "somatic HCN

channels in human neurons can be seen as an evolutionary adaptation that facilitates

otherwise compromised signaling speed in circuits" may be a little simplistic.

Minor comments:

Abstract:

Page 2:

"… precisely timed inhibitory outputs that regulate within neural networks." - missing word (or just regulate neural networks)

"… jitter in the input-output function arise mainly during the transformation of excitatory

postsynaptic potential (EPSP) into action potential output." - should EPSP be plural (EPSPs)?

Page 10: 

"Of the four mouse cells; however, only showed a significant increase in delay" - word missing

"Notably, we found a strong correlation between ZD7288-evoked lengthening of the spike delay and the somal membrane time constant" - somatic membrane time constant

Page 11:

"These so-call basket cells" - so-called

Page 12:

"Somatic HCN channel activity is a common characteristic of these neurons in human but …" - humans

"Despite being called fast-spiking neurons, basket cells in human neocortex actually exhibit…" - in the human neocortex

Reviewer #3: The paper by Szegedi et al. examines the effects of HCN channels on input-output transformation in fast spiking, GABAergic interneurons in the human brain. To address this question, the authors use electrophysiology, superresolution imaging, and modeling. The main findings are:

- Fast-spiking parvalbumin-expressing interneurons show a high density of HCN channels in the perisomatic region in humans.

- Somatic HCN channels enhance input-output fidelity, accelerate somatic membrane potential kinetics, and shorten the delay of action potential generation.

- In human interneurons, somatic leak conductance is low and Ih conductance is high, whereas in mice the situation is opposite. 

Based on these results, the authors conclude that the high perisomatic density of HCN channels in fast-spiking parvalbumin-expressing interneurons in human neocortex contribute to the fast signaling properties of these neurons. Overall, this is a nice paper. As very little information is available about GABAergic interneurons in humans, the manuscript could provide an important contribution to the scientific literature. However, it is also clear that major revision is required before the manuscript can be published in Plos Biology, including both additional experiments and major revision of the text. Given that this is a revised manuscript, I was surprised that parts of the paper were quite preliminary. 

Major points:

1. The immunocytochemical analysis is preliminary and needs to be extended. First, the immunosignals shown in figure 2 need to be quantified, in terms of both number of labeled cells and labeling intensity. Second, the authors focus on HCN1, which is only one out of four potential subunits of HCN channels. HCN2 is also highly expressed in parvalbumin-positive, fast-spiking interneurons (Aponte et al., 2006, J. Physiol.), and some hippocampal interneurons are also immunopositive for HCN3 (Notomi and Shigemoto, 2004, JCN). Finally, the authors describe the expression of Ih as a black and white difference between rodents and humans. However, the data are more consistent with a quantitative difference. The statements needs to be attenuated. 

2. The authors interpret the findings as the difference in the somatic HCN channel density between human and mouse interneurons. However, differences in morphological properties are insufficiently considered. In the extreme case, it may be possible that the conductance densities in the different compartments were the same, and only the different morphological properties would explain the observations (see Mainen and Sejnowski, 1996, Nature, for the marked - indirect - effects of morphology on firing pattern). Cable properties of fast-spiking parvalbumin-expressing interneurons in humans should be used to test this hypothesis. 

3. The cable modeling of the human parvalbumin-expressing interneurons needs to be improved. If the authors want to make the point that - somatic - Ih channels are important, they need to perform simulations with realistic morphology including dendrites, somata, and axon. Furthermore, it is not quite clear what exactly the authors did in their simulations. In the text on page 10, the authors state that they used a single-compartment model. In contrast, in the methods section on page 25, the authors mention that the model was derived from a compartmental model neuron including soma and initial axon segment. Furthermore, which program was used for solving the differential equations? Finally, what was the size of the time steps and the accuracy of the simulations? 

4. The measurement of "membrane time constant" is not entirely correct. In a compartmental model, the membrane time constant is the - slowest - time constant in a multi-exponential waveform. If the authors want to maintain their analysis, they should denote the membrane time constant as "apparent membrane time constant". 

5. The authors claim that ZD 7288 application causes a hyperpolarization of resting membrane potential in human, but not in mouse interneurons (p. 6, top). This is surprising, because ZD 7288 causes the robust hyperpolarization in interneurons in rats (Aponte et al, 2006). The authors should better analyze this difference, also documenting membrane potential as a function of time in figures. Furthermore, the authors should correct misleading statements in the introduction section (page 3, bottom, "small effect of HCN channel blockers on resting membrane potential"). 

6. The action potential phenotype of the recorded human interneurons needs to be better described. Most importantly, both f-I curves and maximal action potential frequency must be given. The authors repeatedly refer to the recorded cells is fast spiking, but the most important criterion (high action potential frequency) is never demonstrated. 

7. The nucleated patch recordings are nice, representing the strongest evidence for the hypothesis that HCN channels are enriched somatically. However, the data shown are incomplete. First, the number of experiments is unclear. Second, a quantitative comparison between mouse and human is missing. Finally, analyzing the properties of nucleated patches in current-clamp conditions is unusual. As the input resistance is ~0.5 GOhm, the seal resistance may contribute to the voltage responses. Voltage-clamp experiments would be preferable, directly revealing the HCN conductance density. 

Minor points:

Page 2, center: "that regulate within neural networks" - unclear. 

Page 5, top: How many nucleated patch recordings were performed? 

Page 6, top: "depolarizing effect" - shouldn't it be "hyperpolarizing"? 

Page 6, center: "in contrast … failed to induce" - in rats slices, ZD 7288 induces a consistent hyperpolarization (Aponte et al., 2006) - see major points above. 

Page 7, bottom: Classical studies suggested the presence of HCN1 immunoreactivity in apical dendrites of neocortical pyramidal neurons (Lörincz et al., 2002, Nature Neuroscience). Clarification is needed here.

Page 9, bottom: "Excitatory currents simulating synaptic excitatory postsynaptic currents EPSCs …" - If this is a dynamic clamp experiment, taking into account changes in driving force, the statement is not quite correct. 

Page 10, top: "for a mean" - unclear. "Only showed" - is it "only one showed"? 

Page 10, center: It is unclear why inwardly rectifying K+ channels were included in the model of the fast-spiking neuron. If available, functional evidence should be cited. 

Page 12, top: "compared rodent basket cells" should read "compared to rodent basket cells". 

Page 12, bottom: "energy expenditure" - it is not easy to see why an Ih current should be more energy-efficient than a leakage potassium current. This argument should be better worked out. 

Page 13, center: "and possible those" should read "and possibly those". It should be possible to figure out all these small annoying mistakes before submission. 

Page 14, top: That the change in the membrane time constant should affect the spike lag is not straightforward. The membrane time constant will primarily determine the decay of the EPSP, whereas the time to peak will be primarily determined by the time course of the AMPA receptor-mediated conductance. 

Page 14, center: That perisomatic leak conductance enhances the fast signaling properties of fast-spiking parvalbumin-positive interneurons has been previously suggested by Noerenberg et al., 2010, PNAS. This aspect should be better discussed in the manuscript.

Page 14, center: Previous work showed that the kinetics of disynaptic inhibition may not only be important for processing speed, but may also be critical to implement novel computations, such as pattern separation (de Almeida et al., 2009, J. Neuroscience). This aspect works in favor of the author's argument and may be added. 

Page 14, bottom: "HCN channel activity can be rapidly up- or downmodulated through intracellular calcium-dependent processes" - this is misleading - the classical intracellular second messenger regulating HCN channels is cAMP. 

Page 19, top: Unspecific wording like "in general" should be removed. 

Page 23, bottom: What is "lg / kg" ? 

Page 24, top: It is unclear why the Cre line should express tdTomato. Was there a crossing with a reporter line (Ai9 or Ai14)? 

Page 24, bottom: "effect of access resistance… was calculated and corrected during data analysis" - if the authors want to achieve an accurate correction, they need to perform series resistance compensation during the experiment at the amplifier level. 

Page 25, center: If recordings are made at physiological temperature, as done here, 3 ms decay time constant is too slow for AMPA receptor-mediated EPSCs in parvalbumin-expressing interneurons.

Page 25, bottom: "in addition to HCN channels … h current" - unclear - isn't the h current the correlate of the HCN channels? 

Page 26: Equations should be numbered for clarity. 

Page 26 / 27: Versions of programs used should be specified. 

Page 28, top: Numerical aperture of objective should be mentioned.

Page 28, center: As parvalbumin immunoreactivity will be washed out during patch-clamp measurements, recording times and access resistance should be quantified in the methods section.

Legend figure 1: "hydrocephalus" is misspelled. 

Legends throughout: The authors should make clear at which time point after ZD application measurements were taken. 

Throughout the paper, the authors refer to patches as outside-out patches, macropatches, or nucleated patches. A more consistent nomenclature should be chosen (previous papers widely called them nucleated patches).

---

## [Decision Letter · Decision Letter 2]

5 Jan 2023

Dear Dr Lämsä,

Thank you for your patience while we considered your further revised manuscript "HCN channels at the somatic membrane ensure rapid input–output function of human neocortex fast-spiking interneurons" for publication as a Research Article at PLOS Biology. This revised version of your manuscript has been evaluated by the PLOS Biology editors, the Academic Editor and the prior reviewers of this iteration of the manuscript. I am happy to say that the reviewers and Academic Editor are now happy with the revisions and we are likely to accept this manuscript for publication. 

At this stage, we simply need you to address some data and other policy-related requests (listed at the bottom of this email). We'd also ask you to consider a slight title change to make the work more accessible to our broad readership. We'd suggest:

HCN channels at the soma of fast-spiking interneurons ensure rapid electrical reactivity in the human neocortex

OR

HCN channels at the cell soma ensure the rapid electrical reactivity of fast-spiking interneurons in human neocortex

OR

The human neocortex requires HCN channels at the soma of fast-spiking interneurons to support rapid electrical reactivity

As you address the data and policy items below, please take this last chance to review your reference list to ensure that it is complete and correct. If you have cited papers that have been retracted, please include the rationale for doing so in the manuscript text, or remove these references and replace them with relevant current references. Any changes to the reference list should be mentioned in the cover letter that accompanies your revised manuscript.

We expect to receive your revised manuscript within two weeks. 

*Published Peer Review History*

*Press*

Sincerely,

Kris

Kris Dickson, Ph.D., (she/her)

Neurosciences Senior Editor/Section Manager,

kdickson@plos.org,

PLOS Biology

DATA POLICY:

Thank you for complying with the PLOS Data Policy, which requires that all data be made available without restriction: http://journals.plos.org/plosbiology/s/data-availability. For more information, please also see this editorial: http://dx.doi.org/10.1371/journal.pbio.1001797. Note that we do not require all raw data. Rather, we ask that all individual quantitative observations that underlie the data summarized in the figures and results of your paper be made available in one of the following forms:

***We have one additional request:

Please ensure that figure legends in your manuscript include information on where the underlying data can be found (e.g. “The underlying data supporting Fig X, panel Y can be found in file Z.”)., and that your supplemental data file/s has a legend.

DATA NOT SHOWN?

- Please note that per journal policy, we do not allow the mention of "data not shown", "personal communication", "manuscript in preparation" or other references to data that is not publicly available or contained within this manuscript. Please check your study carefully for any such statements and either remove mention of such data or provide figures presenting the results and the data underlying the figure(s).

Reviewer remarks:

Do you want your identity to be public for this peer review?

Reviewer #1: No

Reviewer #2: No

Reviewer #3: No

Reviewer #1: I am satisfied with the revision. 

Reviewer #2: The authors have done a very thorough job of addressing all of the critiques of the previous review. The manuscript now makes an important contribution to our understanding of how differences in HCN channel expression contribute to the electrophysiological properties of a defined cell type in mouse versus humans. 

Reviewer #3: The authors addressed the majority of my comments, which significantly improved the paper. From my point of view, the manuscript is suitable for publication in Plos Biol.

---

## [Editor Report · Decision Letter 3]

17 Jan 2023

Dear Dr Lämsä,

Thank you for the submission of your revised Research Article "HCN channels at the cell soma ensure the rapid electrical reactivity of fast-spiking interneurons in human neocortex" for publication in PLOS Biology. On behalf of my colleagues and the Academic Editor, [**AE Name**], I am pleased to say that we can in principle accept your manuscript for publication, provided you address any remaining formatting and reporting issues. These will be detailed in an email you should receive within 2-3 business days from our colleagues in the journal operations team; no action is required from you until then. Please note that we will not be able to formally accept your manuscript and schedule it for publication until you have completed any requested changes.

PRESS

Sincerely, 

Kris Dickson, Ph.D., (she/her)

Neurosciences Senior Editor/Section Manager

PLOS Biology

kdickson@plos.org